# Latent Mamba Operator for Partial Differential Equations

**Karn Tiwari** [1]  **Niladri Dutta** [2]  **N M Anoop Krishnan** [3][4]  **Prathosh A P** [1]

## Abstract

Neural operators have emerged as powerful data-driven frameworks for solving Partial Differential Equations (PDEs), offering significant speedups over numerical methods. However, existing neural operators struggle with scalability in high-dimensional spaces, incur high computational costs, and face challenges in capturing continuous and long-range dependencies in PDE dynamics. To address these limitations, we introduce the Latent Mamba Operator (LaMO), which integrates the efficiency of state-space models (SSMs) in latent space with the expressive power of kernel integral formulations in neural operators. We also establish a theoretical connection between state-space models (SSMs) and the kernel integral of neural operators. Extensive experiments across diverse PDE benchmarks on regular grids, structured meshes, and point clouds covering solid and fluid physics datasets, LaMOs achieve consistent state-of-the-art (SOTA) performance, with a 32.3% improvement over existing baselines in solution operator approximation, highlighting its efficacy in modeling complex PDE solutions. Our code implementation is available at https://github.com/M3RG-IITD/LaMO.

Continuum models describing physical systems are formulated as PDEs across various disciplines, including physics, chemistry, fluid mechanics, and robotics (Debnath & Debnath, 2005). Traditionally, these PDEs are solved by classical numerical methods, such as finite element and spectral methods (Šolín, 2005; Costa, 2004). However, these approaches face significant computational challenges, are elusive to real-time forecasting, are limited in handling diverse boundary conditions, and often require coarse grids for stable solutions (El-metwaly & Kamal, 2024). Additionally, many PDEs, including the Navier-Stokes equations, lack closed-form solutions, posing significant challenges for real-time weather modeling and robotics applications.

In recent years, scientific machine learning (SciML) has introduced neural operators as a promising alternative for solving PDEs through data-driven approaches. Neural operators, an extension of neural networks, have emerged as a powerful tool for mapping between infinite-dimensional functional spaces, serving as universal functional approximations (Li et al., 2020). Unlike traditional methods, neural operators require no prior knowledge of the underlying PDEs. Instead, they rely on data-driven training, enabling faster inference and making them a compelling choice for real-time and complex applications. Neural operators have demonstrated remarkable success across diverse applications, including weather forecasting (Kurth et al., 2023; Bedi et al., 2025), biomedical surrogate modeling (Guan et al., 2021), accelerating sampling processes in diffusion models (Zheng et al., 2023), and as foundation models for solving PDEs (Hao et al., 2024; Herde et al., 2024).

Recent advances in neural operators for solving PDEs have been propelled by transformer-based architectures, with models such as GNOT (Hao et al., 2023), ONO (Xiao et al., 2023), and Transolver (Wu et al., 2024) achieving SOTA performance across diverse PDE tasks. Despite their success, these methods face critical challenges in computational efficiency and scalability, primarily when applied to high-dimensional PDEs. In such cases, where they struggle to parameterize kernel integral transforms (Guibas et al., 2021), leading to overfitting, which limits their generalization. A fundamental bottleneck lies in the quadratic complexity of the self-attention mechanism inherent to conventional transformers, which incurs prohibitive computational costs, impedes the processing of continuous signals, and degrades inference speed as spatial resolution increases (Hao et al., 2023; Wu et al., 2024).

While recent efforts, such as Galerkin-type attention (Cao, 2021), address the issue of computational scaling by reducing the complexity to linear, this compromise significantly

---

[1]Department of Electrical Communication Engineering, Indian Institute of Science, Bangalore, India [2]Department of Computer Science and Automation, Indian Institute of Science, Bangalore, India [3]Yardi School of AI, Indian Institute of Technology, New Delhi, India [4]Department of Civil Engineering, Indian Institute of Technology, New Delhi, India. Correspondence to: Karn Tiwari <karntiwari@iisc.ac.in>, N M Anoop Krishnan <krishnan@iitd.ac.in>, Prathosh A P <prathosh@iisc.ac.in>.

*Proceedings of the 42$^{nd}$ International Conference on Machine Learning*, Vancouver, Canada. PMLR 267, 2025. Copyright 2025 by the author(s).

diminishes the expressive power of the self-attention mechanism, limiting the ability to capture intricate dynamical interactions in complex PDE systems. Similarly, global convolution-based paradigms like FNOs (Li et al., 2020) leverage spectral convolution to encode global information but encounter persistent challenges, including spectral decay, vanishing gradients, and computational bottlenecks when processing high-resolution inputs (Tran et al., 2021; Fanaskov & Oseledets, 2022). These limitations underscore a pressing need for a framework that balances expressive power, computational efficiency, and scalability in PDE learning, which can generalize better with limited data.

Structured state-space models (SSMs) have emerged as a promising approach for sequence models, demonstrating excellent performance in language processing. These models offer linear scaling with improved long-range dependency modeling, efficient memory usage, and superior handling of language and vision data (Gu & Dao, 2023; Zhu et al., 2024). However, the application of such a framework toward PDEs remains poorly explored. Here, we propose SSMs in operator learning and introduce a unified framework, Latent Mamba Operator (LaMO). LaMO leverages latent space and bidirectional SSMs to address the limitations of existing Neural Operators, providing an efficient, scalable, and effective solution for complex tasks in SciML. The main contributions of our work are briefly summarized below.

1. **LaMO**: We propose **Latent Mamba Operator (LaMO)**. This framework leverages a latent physical space combined with bidirectional state-space models (SSMs) to learn underlying solution operators.

2. **Theoretical Insights**: We provide a theoretical analysis of LaMO from the operator's perspective, demonstrating its capability to act as a kernel integral for learning underlying solution operators in Theorem 3.4.

3. **Superior Performance**: Extensive experiments are conducted across various regular and irregular grid datasets, showcasing an average improvement of 32.3% over the SOTA models as presented in Table 1.

## 1. Related Work

### 1.1. Neural Operator (NO)

Neural operators have demonstrated significant success in solving parametric PDEs through data-driven approaches (Kovachki et al., 2021). Lu et al. (2021) introduced DeepONet, which established the universal functional approximation capability of neural operators. DeepONet employs two networks: the branch network, responsible for learning the input function operator, and the trunk network, which projects this operator onto the target function space. Another prominent neural operator, the FNO (Li et al., 2020),

leverages frequency domain techniques. FNO employs Fourier kernel-based integral transformations facilitated by fast Fourier transform and projection blocks. An enhanced version, F-FNO (Tran et al., 2021), improves upon FNO by incorporating advanced spectral mixing and residual connections. Various kernel integral neural operators have been proposed based on frequency kernel methods. For instance, Fanaskov & Oseledets (2022) introduced spectral methods based on Chebyshev and Fourier series to reduce aliasing errors and improve the clarity of FNO outputs. In contrast, CoNO (Tiwari et al., 2024) employs a Fractional Fourier transform (FrFT)-based integral kernel. Interestingly, Kovachki et al. (2021) showed that the self-attention mechanism can be interpreted as a specific case of neural operators learning an integral kernel for solving PDEs.

### 1.2. Transformer-Based Neural Operators

Recent work has sought to enhance the efficiency of attention-based neural operators. Cao (2021) pioneered this direction by replacing traditional softmax layers with two novel self-attention operators, significantly reducing computational overhead while grounding attention mechanisms in kernel integral theory. Subsequent work by Hao et al. (2023) introduced the GNOT operator, which leverages a linear cross-attention block to improve geometric encoding for irregular domains. Despite these advances, transformer-based operators remain vulnerable to overfitting in data-scarce regimes, limiting their generalization. To mitigate this, Xiao et al. (2023) proposed orthogonal regularization, enhancing robustness by enforcing orthogonality in attention weights. Meanwhile, the SOTA Transolver operator (Wu et al., 2024) addresses scalability challenges in large meshes through latent attention, decoupling computational complexity from mesh resolution.

### 1.3. State Space Models (SSMs)

Structured State Space (S4) models, introduced by (Gu et al., 2021a), provide a novel alternative to transformers for modeling long-range sequences with linear scaling in sequence length, unlike the quadratic complexity of transformers. Gu et al. 2021b highlighted the potential of SSMs for capturing long-range dependencies. Subsequent advancements include complex diagonal structures, low-rank decompositions, and selection mechanisms (Dao & Gu, 2024). Recently, Hu et al. 2024 proposed the integration of SSMs in solving ordinary differential equations (ODEs) for time series datasets, and MemNO (Buitrago et al.) introduced the benefits of incorporating SSMs for time-dependent PDEs. While SSMs have excelled in language (Gu & Dao, 2023; Qu et al., 2024), audio (Erol et al., 2024), and computer vision tasks (Zhu et al., 2024; Liu et al., 2024), their applicability to solving parametric high-dimensional PDEs remains largely unexplored in SciML.

## 2. Preliminaries

### 2.1. Problem Statement

The main objective is to learn the nonlinear mapping between two infinite-dimensional spaces through observed input-output pairs (Li et al., 2020; Lu et al., 2021). Let $D$ represent an open and bounded domain as $D \subset \mathbb{R}^d$, with $A = A(D; \mathbb{R}^{d_a})$ and $U = U(D; \mathbb{R}^{d_u})$ as separable banach spaces of functions representing elements in $\mathbb{R}^{d_a}$ and $\mathbb{R}^{d_u}$, respectively. Let $\mathcal{G}^\dagger : A \to U$ denotes a non-linear mapping arising from the underlying solution operator for parametric PDEs.

In operator learning, we aim to construct an approximation for $\mathcal{G}^\dagger$ via a parametric mapping $\mathcal{G} : A \times \Theta \to U$, or equivalently, $\mathcal{G}_\theta : A \to U, \theta \in \Theta$, within a finite-dimensional parameter space $\Theta$ via i.i.d. observations $(a_j, u_j)_{j=1}^N$, where $a_j \sim \mu$, drawn from the underlying measure $\mu$ supported on $A$, and $u_j = \mathcal{G}^\dagger(a_j)$. The aim is to learn $\theta^\dagger \in \Theta$ such that $\mathcal{G}(\cdot, \theta^\dagger) = \mathcal{G}_\theta^\dagger \approx \mathcal{G}^\dagger$. It allows learning in infinite-dimensional spaces as the solution to the optimization problem in Eq. 1 constructed using a loss function $\mathcal{L} : U \times U \to \mathbb{R}$.

$$\min_{\theta \in \Theta} \mathbb{E}_{a \sim \mu} \left[ \mathcal{L}(\mathcal{G}_\theta(a), \mathcal{G}^\dagger(a)) \right] \quad (1)$$

The optimization problem is solved using a data-driven empirical risk minimization of the loss function, akin to the supervised learning approach using train-test observations.

### 2.2. State Space Models

**SSMs:** Classical SSMs define a continuous system (Linear Time-Invariant (LTI) system) that maps an input sequence $\mathbf{x}(t) \in \mathbb{R}^L$ into a hidden representation $\mathbf{h}(t) \in \mathbb{R}^N$, which is then used to predict an output response $\mathbf{y}(t) \in \mathbb{R}^L$. Formally, an SSMs can be formulated as ordinary differential equations (ODEs) as follows:

$$\mathbf{h}'(t) = \mathbf{A}\mathbf{h}(t) + \mathbf{B}\mathbf{x}(t), \quad (2)$$
$$\mathbf{y}(t) = \mathbf{C}\mathbf{h}(t), \quad (3)$$

where $\mathbf{A} \in \mathbb{R}^{N \times N}$, $\mathbf{B} \in \mathbb{R}^{N \times 1}$, and $\mathbf{C} \in \mathbb{R}^{1 \times N}$ are learnable model parameters.

**Discretization:** To use continuous SSMs for integration into neural networks, it is essential to apply discretization operations. By introducing a timescale parameter $\Delta \in \mathbb{R}$ and using the zero-order hold (ZOH) method as the discretization rule, the discrete counterparts of $\mathbf{A}$ and $\mathbf{B}$ (denoted as $\overline{\mathbf{A}}$ and $\overline{\mathbf{B}}$, respectively). Consequently, Equation 3 can be rephrased into a discretized form as:

$$\mathbf{h}[k] = \overline{\mathbf{A}}\mathbf{h}[k-1] + \overline{\mathbf{B}}\mathbf{x}[k], \quad \mathbf{y}[k] = \mathbf{C}\mathbf{h}[k], \quad (4)$$

where,

$$\overline{\mathbf{A}} = e^{\Delta \mathbf{A}}, \quad \overline{\mathbf{B}} = (\Delta \mathbf{A})^{-1}(e^{\Delta \mathbf{A}} - \mathbf{I})\mathbf{B} \approx \Delta \mathbf{B}, \quad (5)$$

and $\mathbf{I}$ represents the identity matrix. Subsequently, the process in Eq. 5 can be realized in a global convolution form:

$$\mathbf{y} = \mathbf{x} * \mathbf{K}, \quad \mathbf{K} = [\mathbf{CB}, \mathbf{CAB}, \ldots, \mathbf{CA}^{L-1}\mathbf{B}], \quad (6)$$

where $\mathbf{K} \in \mathbb{R}^L$ represents the convolutional kernel.

**Selective State Space Models:** The Selective State Space (S6) mechanism, proposed in Mamba (Gu & Dao, 2023), introduces input-dependence for the parameters $\mathbf{B}$, $\mathbf{C}$, and $\Delta$, significantly improving the performance of SSMs and parallel scanning to make the training performance comparable with transformer-based models. By making these parameters input-dependent, the global convolution kernel in Equation 6 can be reformulated as follows:

$$\mathbf{K} = \{\mathbf{C}_L\mathbf{B}_L, \mathbf{C}_L\mathbf{A}_{L-1}\mathbf{B}_{L-1}, \ldots, \mathbf{C}_L \prod_{i=1}^{L-1} \mathbf{A}_i\mathbf{B}_1\}. \quad (7)$$

The reformulation highlights that integrating selective state-space parameters (data-dependent parameters) enhances the expressiveness of the SSMs for complex tasks. Better efficiency is attained through an IO-aware implementation of associative scans, leveraging work-efficient parallel scanners to enable parallelization on modern hardware. In LaMO, $\mathcal{G}_\theta$ is parameterized using the SSMs as an integral kernel.

*Remark* 2.1. The matrix $\mathbf{A}$ of Mamba has diagonal structure i.e $\mathbb{A} = \text{Diag}(\lambda_1, \ldots, \lambda_p)$ with $\lambda_i \in \mathbb{C} \, \forall i$.

## 3. Proposed Method

Transformers struggle to capture kernel integral transforms efficiently in complex, high-dimensional continuous PDEs (Guibas et al., 2021; Karniadakis et al., 2021). To address this, we introduce LaMO (Latent Mamba Operator), inspired by structured state-space models like Mamba (Gu & Dao, 2023). LaMO begins with a Latent Encoder inspired by the Perceiver (Jaegle et al., 2021), which condenses PDE's physical tokens into a compact latent representation. These tokens are then processed by an SSMs block (bidirectional), utilizing SSM's capability to learn data-dependent kernels, thereby effectively modeling complex PDE dynamics.

### 3.1. Latent Mamba Operator (LaMO) Architecture

**Overview:** The LaMO operator takes the following form:

$$\mathcal{G}_\theta = \mathcal{Q} \circ \mathcal{D} \circ \mathcal{L}^l \circ \ldots \mathcal{L}^2 \circ \mathcal{L}^1 \circ \mathcal{E} \circ \mathcal{P}, \quad (8)$$

where $\circ$ denotes composition. The operators $\mathcal{P} : \mathbb{R}^d \to \mathbb{R}^d$ and $\mathcal{Q} : \mathbb{R}^d \to \mathbb{R}^d$ correspond to the lifting and projection operations, encoding lower-dimensional spaces into higher-dimensional spaces or vice versa, which helps in converting the non-linear dynamics into linear dynamics (Bevanda et al., 2021) using the feed-forward neural network. The operations $\mathcal{E}$ and $\mathcal{D}$ correspond to the encoder and decoder,

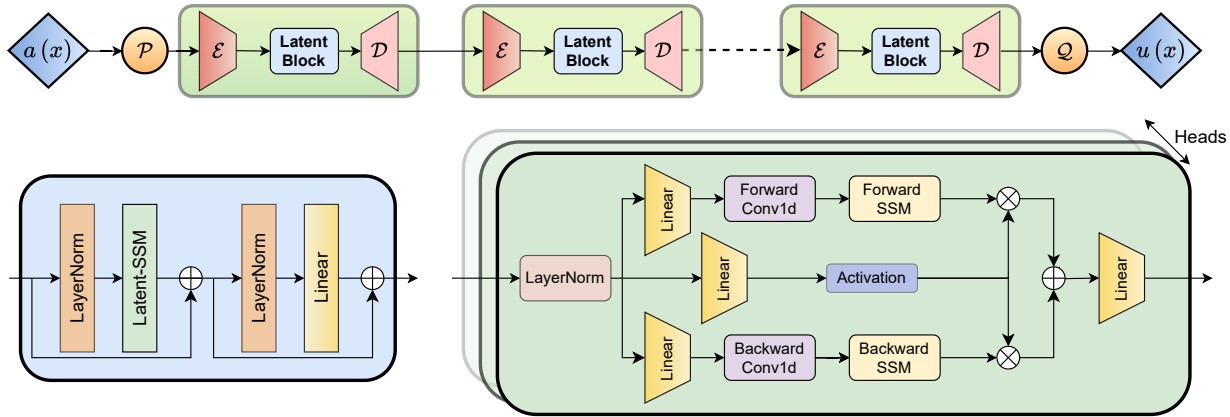

*Figure 1.* **Overview.** (1) The input function $a(x)$ is lifted to a higher-dimensional representation using the lifting operator $\mathcal{P}$. (2) The encoder $\mathcal{E}$ maps the input from the physical domain to the latent domain, where the latent block (**Bottom Left**) performs the kernel integral via SSMs, applies channel mixing and decodes the latent tokens back to the physical domain using the decoder $\mathcal{D}$. (3) A multi-headed bidirectional SSM (**Bottom Right**) is applied across the tokens within the latent SSMs block. (4) This process is repeated $\mathcal{L}$ times. Finally, the channel dimensions are reduced to the desired output size using the projection operator $\mathcal{Q}$, yielding the final output function $u(x)$.

transforming the input domain into latent tokens and vice versa. These operators are constructed using $l$ layers of non-linear integral operators, denoted as $\mathcal{L}^l$, consisting of an integral kernel $\mathcal{K}$, parametrized using the time-variant SSMs. The design of $\mathcal{L}^l$ is inspired by the integral operators (Green's functions) commonly used for solving linear PDEs.

***Encoder*** ($\mathcal{E}$): The encoder module draws inspiration from the design principles of Perceiver (Jaegle et al., 2021), which efficiently reduces the number of tokens for computational tractability. Unlike the Perceiver, where latent tokens correspond to a fixed latent representation, this module derives latent tokens directly from the input physical tokens. Let $\mathbf{X} \in \mathbb{R}^{N \times D}$ denote the input tokens, where $N$ is the number of tokens and $d$ is the embedding dimension. We project physical $N$ tokens into $M$ latent tokens where $M \ll N$ which can be described as follows:

$$\mathbf{W} = \text{Softmax}(\text{Linear}(\mathbf{X})), \quad (9)$$

$$\mathbf{Z} = \mathbf{W}^T \mathbf{X}, \quad (10)$$

where $\text{Linear}()$ projects the $D$ channels into $M$ channels using a feed-forward neural network, $\mathbf{W} \in \mathbb{R}^{N \times M}$ denotes the latent weights which are adaptive as its function of input tokens and $\mathbf{Z}$ denotes the latent tokens which correspond to latent space which in contrast to a frequency-based method such as FNO, we learn the latent space depending on the input tokens. The $\text{Softmax}()$ operation, similar to that used in Transolver and Perceiver, ensures that the latent weights have a low entropy, resulting in an informative latent space. *Remark* 3.1 (**ViT Patches as Special Cases of Latent Tokens** (Alexey, 2020)). The latent encoder operation divides the grid into $M$ equal-sized square patches for regular computational grids. If a point $\mathbf{X}_i$ belongs to the $j$-th

patch, the patchify operation can be approximated by setting $\mathbf{Z} = \mathbf{W}^T \mathbf{X}$ and optimizing latent weights $\mathbf{W}_i$ such that $\mathbf{W}_{i,j} \approx 1$ and $\mathbf{W}_{i,k} \approx 0$ for $k \neq j$. Thus, the patch operation is a special case of latent tokens for regular grids.

***Latent Block:*** Inspired by the attention architecture, we describe the LaMO block as a combination of a latent SSMs operation, which defines the kernel integral operator, followed by channel mixing. Suppose we are using $L$ layers; the $l$-th layer of the LaMO block can be expressed as:

$$\hat{\mathbf{Z}}^l = \mathbf{Z}^{l-1} + \text{Latent-SSM}\left(\text{LayerNorm}\left(\mathbf{Z}^{l-1}\right)\right), \quad (11)$$

$$\mathbf{Z}^l = \hat{\mathbf{Z}}^l + \text{Linear}\left(\text{LayerNorm}\left(\hat{\mathbf{Z}}^l\right)\right), \quad (12)$$

where $l \in \{1, \cdots, L\}$, $\text{Linear}()$ represent a feed-forward neural network and $\text{LayerNorm}()$ represent the layer normalization (Lei Ba et al., 2016). Here, $\mathbf{Z}^l \in \mathbb{R}^{M \times D}$ denotes the output of the $l$-th layer, and $\mathbf{Z}^0 \in \mathbb{R}^{M \times D}$ represents the latent tokens embedded from the input physical tokens.

***Latent SSM Block:*** SSMs are primarily designed for 1-dimensional sequences, typically operating in a causal manner, where the output at any step depends on the arrangement of preceding tokens. However, accounting for the entire sequence in a non-causal setting for tasks involving PDEs is essential. To address this, we use the bidirectional SSM block, which leverages both forward and backward kernels to process the sequence holistically.

The appendix Algorithm 1 provides the algorithm detailing the operations performed. Specifically, the latent token

sequence $\mathbf{Z}$ undergoes following operation as follows:

$$
\begin{aligned}
\hat{\mathbf{X}} &= \text{Linear}(\mathbf{Z}), \\
\hat{\mathbf{Z}} &= \text{Linear}(\mathbf{Z}), \\
(\overline{\mathbf{A}}, \overline{\mathbf{B}}, \mathbf{C}) &\leftarrow \text{Discretization}(\hat{\mathbf{X}}), \\
\mathbf{Y} &= \sum_{\text{d} \in \text{Direction}} \text{SSM}(\mathbf{X}_{\text{d}}) \otimes \text{Act}(\hat{\mathbf{Z}}), \\
\mathbf{Z} &= \text{Linear}(\mathbf{Y}).
\end{aligned}
\tag{13}
$$

where $\text{Linear}()$ represents a feed-forward neural network, while direction refers to the SSM recurrence applied in the forward and backward directions. The activation function $\text{Act}()$ is implemented as SiLU (Elfwing et al., 2018), and $\text{Discretization}()$ denotes the process of converting continuous dynamics into discrete dynamics using the zero-order hold (ZOH). Finally, the block's output is obtained by applying a gating mechanism followed by a linear projection.

*Remark* 3.2 (**Multidirectional Scan on Regular Grid**). For latent tokens on a regular grid, the spatial arrangement of tokens is crucial due to the inherent continuity in regular grids. Unlike irregular grids, we utilize a multidirectional scan along four paths: (i) top-left to bottom-right, (ii) bottom-right to top-left, (iii) top-right to bottom-left, and (iv) bottom-left to top-right. Each direction is processed in parallel, reducing the overall computational complexity to linear, and has proven effective for regular grids. The multidirectional scan is further analyzed and validated experimentally in the appendix Section E.1.

***Decoder*** ($\mathcal{D}$): Decoder can be defined as converting the latent tokens back to physical tokens. We need to convert the latent token $\mathbf{Z} \in \mathbb{R}^{M \times D}$ into the physical tokens $\mathbf{Y} \in \mathbb{R}^{N \times D}$ where $M \ll N$. Formally, it can be described as:

$$
\mathbf{W} = \text{Softmax}(\text{Linear}(\mathbf{Z})), \tag{14}
$$

$$
\mathbf{Y} = \mathbf{W}^T \mathbf{Z}, \tag{15}
$$

where $\text{Linear}()$ projects the $D$ channels into $N$ channels using a feed-forward neural network, $\mathbf{W} \in \mathbb{R}^{M \times N}$ denotes the latent weights and $\text{Softmax}()$ operation, ensures that the latent weights have a low entropy similar to the encoder.

***Multihead:*** To enhance the model's effectiveness, we employ a multihead architecture similar to the attention mechanism (Vaswani, 2017) that captures information from different representational subspaces. Specifically, a latent token embedding dimension $D$ is split into $H$ heads, each with a dimensionality of $D/H$. The latent tokens are processed as:

$$
\text{SSM}(\mathbf{Z}) = \text{Concat}(\text{SSM}(\mathbf{Z}_l^1), \dots, \text{SSM}(\mathbf{Z}_l^H)). \tag{16}
$$

To integrate features from different heads, the outputs of all heads are concatenated and linearly projected, yielding the final output. This architecture increases model diversity while maintaining a compact size. Furthermore, our experiments demonstrate that the multi-head mechanism accelerates training convergence and reduces over-fitting.

***Shared Weights:*** The transformation between latent and physical tokens allows LaMO to build deeper models but may limit its ability to fully capture physical space information due to increased computational complexity. We share encoder and decoder parameters between blocks to balance performance and efficiency (Jaegle et al., 2021).

***Computational Analysis:*** For simplicity, the above process is summarized as $\mathbf{Z}' = \text{Latent-SSM}(\mathbf{Z})$ with an overall computational complexity of $\mathcal{O}(NMD + MD)$. Since $M$ is set as a constant and $M \ll N$, the computational complexity becomes linear with respect to the number of mesh points. Keeping the mesh size constant, it is also linear with respect to the number of latent tokens.

### 3.2. Theoretical Analysis

In this subsection, we present a theoretical analysis of the components of the proposed method. Specifically, (i) we establish the equivalence between the SSM and the Euler method for solving PDEs, which is essential for understanding the performance of SSM (Proposition 3.3). (ii) Next, we prove that SSM serves as a Monte Carlo approximation of a learnable integral kernel (appendix Lemma B.10). This result is then used to prove that the latent SSM is equivalent to a learnable kernel integral in Theorem 3.4. appendix Section B provides a more detailed analysis.

**Proposition 3.3.** *The Zero-Order Hold (ZOH) discretization of continuous parameters is expressed as follows:*

$$
\begin{aligned}
\overline{\mathbf{A}} &= \exp(\Delta \mathbf{A}), \\
\overline{\mathbf{B}} &= (\Delta \mathbf{A})^{-1} \big( \exp(\Delta \mathbf{A}) - \mathbf{I} \big) \cdot \mathbf{B},
\end{aligned}
\tag{17}
$$

*where $\Delta$ denotes the time step, $\mathbf{I}$ is the identity matrix, and $\mathbf{A}, \mathbf{B}$ are continuous system matrices. The discretization method aligns with the Euler method, approximating the matrix exponential by truncating its Taylor series expansion to the first-order term.*

The complete proof is in the appendix Section B.4. The above proposition establishes the equivalence between the continuous and discrete SSM parameters and their relation to the Euler method for solving the PDE. Understanding SSM's performance from the Euler method's perspective is crucial. The ZOH discretization method preserves higher-order terms, making it more accurate for solving PDEs.

Next, we demonstrate that SSM functions as an integral kernel, following prior works (Kovachki et al., 2021; Cao, 2021). The Lemma B.10 shows that the canonical SSM is the Monte Carlo approximation of the integral kernel, used iteratively to solve the solution operator. However, we apply the kernel integral on latent tokens. For better understanding,

*Table 1.* The main results across all benchmark datasets are presented using the mean relative $\ell_2$ error (Equation 21) as the evaluation metric. A lower $\ell_2$ error signifies better performance. The "INCREMENT %" quantifies the relative error reduction achieved by our model compared to the second-best performer on each benchmark. For clarity, in the color legend, orange represents the best performance, blue indicates the second-best performance, and violet signifies the third-best performance among the baselines.

| | OPERATOR | POINT CLOUD | REGULAR GRID | | STRUCTURED MESH | | |
| | | ELASTICITY | NAVIER–STOKES | DARCY | PLASTICITY | AIRFOIL | PIPE |
|---|---|---|---|---|---|---|---|
| **CLASSIC** | UNET (2015) | 0.0235 | 0.1982 | 0.0080 | 0.0051 | 0.0079 | 0.0065 |
| | RESNET (2016) | 0.0262 | 0.2753 | 0.0587 | 0.0233 | 0.0391 | 0.0120 |
| | SWIN (2021) | 0.0283 | 0.2248 | 0.0397 | 0.0170 | 0.0270 | 0.0109 |
| | DEEPONET (2021) | 0.0965 | 0.2972 | 0.0588 | 0.0135 | 0.0385 | 0.0097 |
| **FREQUENCY** | WMT (2021) | 0.0359 | 0.1541 | 0.0082 | 0.0076 | 0.0075 | 0.0077 |
| | U-FNO (2022) | 0.0239 | 0.2231 | 0.0183 | 0.0039 | 0.0269 | 0.0056 |
| | FNO (2020) | 0.0229 | 0.1556 | 0.0108 | 0.0074 | 0.0138 | 0.0067 |
| | U-NO (2022) | 0.0258 | 0.1713 | 0.0113 | 0.0034 | 0.0078 | 0.0100 |
| | F-FNO (2021) | 0.0263 | 0.2322 | 0.0077 | 0.0047 | 0.0078 | 0.0070 |
| | LSM (2023) | 0.0218 | 0.1535 | 0.0065 | 0.0025 | 0.0059 | 0.0050 |
| **TRANSFORMER** | GALERKIN (2021) | 0.0240 | 0.1401 | 0.0084 | 0.0120 | 0.0118 | 0.0098 |
| | HT-NET (2022) | / | 0.1847 | 0.0079 | 0.0333 | 0.0065 | 0.0059 |
| | OFORMER (2022B) | 0.0183 | 0.1705 | 0.0124 | 0.0017 | 0.0183 | 0.0168 |
| | GNOT (2023) | 0.0086 | 0.1380 | 0.0105 | 0.0336 | 0.0076 | 0.0047 |
| | FACTFORMER (2024) | / | 0.1214 | 0.0109 | 0.0312 | 0.0071 | 0.0060 |
| | ONO (2023) | 0.0118 | 0.1195 | 0.0076 | 0.0048 | 0.0061 | 0.0052 |
| | TRANSOLVER (2024) | 0.0064 | 0.0957 | 0.0059 | 0.0013 | 0.0053 | 0.0046 |
| **SSM** | **LaMO (OURS)** | **0.0050** | **0.0460** | **0.0039** | **0.0007** | **0.0041** | **0.0038** |
| | INCREMENT % | **21.8%** | **51.9%** | **33.9%** | **46.1%** | **22.6%** | **17.4%** |

we show that the latent SSM in our framework is a learnable kernel integral in the latent domain in Theorem 3.4.

**Theorem 3.4** (**SSM as an equivalent integral kernel on** $\Omega$). *Let $\Omega \subseteq \mathbb{R}^n$ be a bounded domain, $\boldsymbol{a} : \Omega \to \mathbb{R}^d$ be a given input function, and $\mathbf{x} \in \Omega$ be a mesh point. An latent-SSM layer approximates the integral operator $\mathcal{G} : L^2(\Omega, \mathbb{R}^d) \to L^2(\Omega, \mathbb{R}^d)$, defined as follows:*

$$\mathcal{G}(\boldsymbol{a})(\mathbf{x}) = \int_\Omega \kappa(\mathbf{x}, y)\, \boldsymbol{a}(y)\, \mathrm{d}y, \qquad (18)$$

*where $\kappa : \Omega \times \Omega \to \mathbb{R}^{d \times d}$ is the kernel function characterizing the operator $\mathcal{G}$.*

The appendix Theorem B.12 provides the complete proof. In prior work, (Li et al., 2020) formalized the neural operator as an iterative process, where the key component is a linear kernel integral. This formulation consists of an integral operator combined with a nonlinear activation function, which enables the learning of a nonlinear surrogate mapping. Since the nonlinear activation function can be efficiently parametrized using a feedforward neural network, the above theorem demonstrates that the SSM can be viewed as a linear kernel integral. This result implies that an SSM-based operator can effectively learn the nonlinear surrogate mapping for the underlying PDE solution. Now, we show the connection of the proposed method with the recent transformer-based baseline ONO (Xiao et al., 2023).

**Connection with ONO** (Xiao et al., 2023): ONO introduces orthogonal attention to model the kernel update using Cholesky Decomposition (Higham, 1990), ensuring the orthogonality and positive definiteness of the attention kernel. The orthogonal attention kernel was defined as follows:

$$\kappa(x, y) = \psi'(x)\mathrm{Diag}(\mu)\psi(y), \qquad (19)$$

where $\psi(\cdot) : \Omega \to \mathbb{R}^d$ maps the input to an embedding space. As shown in the appendix Section B, this structure resembles the inherent form of the SSM kernel:

$$\kappa(x, y) = C(x)\mathrm{Diag}\left(\prod A\right) B(y). \qquad (20)$$

By leveraging this inherent structure, LaMO avoids explicit decomposition for orthonormalization, thereby mitigating over-fitting and improving generalization.

## 4. Numerical Experiments

This section presents a comprehensive empirical evaluation of LaMO against several neural operator baselines. We conduct extensive experiments on diverse standard benchmarks to validate the proposed method.

### 4.1. Implementation Details

***Benchmark:*** We evaluate LaMO's performance on regular grids using the Darcy and Navier-Stokes (Li et al.,

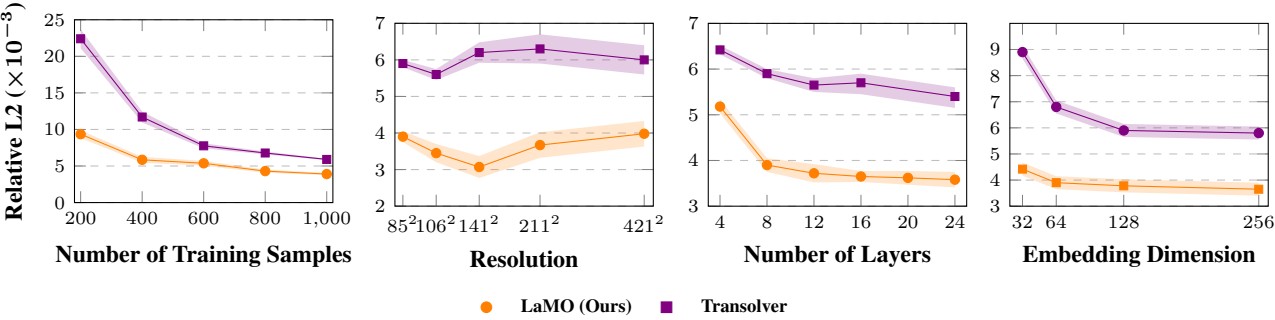

*Figure 2.* Model performance on the scalability of the Darcy Flow benchmark evaluated across various aspects: (**Left**) Data Efficiency, measuring performance with varying amounts of training data; (**Middle Left**) Resolution, assessing the impact of different input spatial resolutions; (**Middle Right**) Model Depth, analyzing performance with increasing layers; and (**Right**) Embedding Dimension, examining the effect of varying latent space dimensionality. Lower relative $l_2$ error ($\times 10^{-3}$) indicates better performance.

2020) benchmarks. We then extend our experiment to irregular geometries, including Airfoil, Plasticity, and Pipe (Li et al., 2022a) with structured meshes and Elasticity (Li et al., 2022a) represented as point clouds. Further details are provided in the appendix Section C.

***Baselines:*** We evaluate LaMO against over 15+ baselines, ranging from frequency-based operators such as FNO (Li et al., 2020), U-FNO (Wen et al., 2022), WMT (Gupta et al., 2021), F-FNO (Tran et al., 2021), U-NO (Rahman et al., 2022) to transformer-based operators such as GalerkinTransformer (Cao, 2021), HT-Net (Liu et al., 2022), GNOT (Hao et al., 2023), Factformer (Li et al., 2024), OFormer (Li et al., 2022b), ONO (Xiao et al., 2023), and Transolver (Wu et al., 2024), representing SOTA neural operator.

***Evaluation Metric (Relative $l_2$)*** (Li et al., 2020): Mean Relative $\ell_2$ error is used as metric throughout the experiments.

$$\mathcal{L} = \frac{1}{N} \sum_{i=1}^{N} \frac{\|\mathcal{G}_\theta(a_i) - \mathcal{G}^\dagger(a_i)\|_2}{\|\mathcal{G}^\dagger(a_i)\|_2} \qquad (21)$$

The regular mean-squared error (MSE) is enhanced with a normalizer $\|\mathcal{G}^\dagger(a_i)\|_2$ to take account of discrepancies in absolute resolution scale across different benchmarks.

***Implementation Details:*** We use the mean relative $\ell_2$ error (Equation 21) as the training and evaluation metric. All models are trained for 500 epochs with the AdamW (Loshchilov et al., 2017) optimizer and OneCycleLR scheduler (Smith & Topin, 2019). To ensure a fair comparison, we have kept our model parameters equal to or fewer than those of the transformer-based baselines. Further details on implementation and hyperparameters are provided in the appendix. Experiments are conducted on a Linux machine with Ubuntu 20.04.3 LTS, an Intel(R) Core(TM) i9-10900X processor, and a single NVIDIA A100 40 GB GPU. More comprehensive details are available in the appendix Section D.

### 4.2. Main Results

We benchmark LaMO against standard baselines and various latest SOTA neural operators (Wu et al., 2023; 2024; Hao et al., 2023), ranging from frequency-based to transformer-based neural operators. Table 1 shows that LaMO outperforms these baselines across diverse physics domains, including solid and fluid dynamics on various geometries. On average, LaMO achieves a 32.3% improvement over the second-best baseline, with a remarkable 49% gain on time-dependent PDEs like Navier-Stokes ($0.095 \rightarrow 0.046$) and Plasticity ($0.0013 \rightarrow 0.0007$) benchmark. It demonstrates LaMO's superior ability to handle complex dynamics compared to transformer-based approaches. While transformer-based models such as GNOT and Oformer apply attention directly to mesh points, they struggle to capture complex fluid interactions, as evidenced by the Darcy and turbulent Navier-Stokes flows (Foias et al., 2001; Bungartz & Schäfer, 2006). In contrast, LaMO's use of latent tokens and SSM proves more effective for modeling the intricate dynamics of regular fluid datasets, despite Transolver addressing some of these issues with physics-informed attention. These results highlight the effectiveness of our framework in operator learning for learning accurate surrogate solution approximation.

### 4.3. Ablation Results

***Ablations:*** In addition to the main result, we carry out the ablations of the components in LaMO. Table 2 highlights the impact of SSM's varying numbers of latent tokens and state dimensions (DState) on model performance. Increasing the token number consistently yields lower errors, indicating that finer partitioning provides finer input representation. Increasing DState dimensions generally improves performance, particularly with larger latent token sizes, though it shows diminishing returns with a decrease in latent tokens. The best performance (0.0038) on Darcy is achieved

with increasing the latent tokens and (DState = 64) with expand dim = 2, underscoring the importance of balancing fine-grained latent tokens and is crucial for achieving better performance with balanced computation.

*Table 2.* Ablation Study on a Number of Latent Tokens and SSM Parameters: We evaluate three variants on the Darcy, varying the latent token number, SSM dstate, and SSM expanding dimensions. Lower relative $l_2$ error indicates better model performance.

| | | RELATIVE L2 LOSS | | | |
|---|---|---|---|---|---|
| | LATENT TOKENS | STATE DIMENSIONS OF SSM | | | |
| | | 1 | 16 | 32 | 64 |
| **EXPAND DIM** **1** | **1936** | 0.0045 | 0.0044 | 0.0041 | 0.0040 |
| | 484 | 0.0053 | 0.0052 | 0.0047 | 0.0056 |
| | 121 | 0.0068 | 0.0065 | 0.0070 | 0.0071 |
| **2** | **1936** | 0.0041 | 0.0042 | 0.0039 | 0.0038 |
| | 484 | 0.0049 | 0.0048 | 0.0051 | 0.0047 |
| | 121 | 0.0065 | 0.0067 | 0.0066 | 0.0066 |

***Ablation Study on Bidirectional SSM:*** To further assess the necessity of bidirectional SSM, we conducted experiments with unidirectional SSM, which failed to perform consistently across benchmarks (on Plasticity, it becomes worse as $0.0007 \rightarrow 0.0022$). Thus confirming the significance of capturing dynamics in a non-causal setting. More Ablation can be found in the appendix, Subsection E.1.

***Effect of Non-Periodic Boundary Condition***: LaMO performs well on challenging non-periodic boundary benchmarks, including Darcy flow and Navier-Stokes, as evidenced by the results in Table 1. It demonstrates its robustness in handling complex real-world scenarios where traditional periodic boundary assumptions fail.

***Effect of the Number of Latent Tokens***: As shown in Table 3, we systematically increase the number of latent tokens in the Airfoil benchmark. The results indicate that a higher number of latent tokens allows both Transolver and LaMO to capture finer details in the physical space, albeit at the cost of increased computation. Moreover, performance initially improves with an increasing number of latent tokens. Eventually, it deteriorates beyond a certain threshold, suggesting the existence of an optimal number of latent tokens that balances accuracy and computational efficiency. LaMO consistently outperforms Transolver across all tested latent token configurations, demonstrating its robustness and superior capability in capturing underlying physical dynamics using LaMO.

### 4.4. Model Analysis

***Scaling:*** To further investigate the scalability of LaMO as a potential foundation model for PDE solvers, we analyzed its performance under varying training data samples, res-

*Table 3.* Ablation on the number of latent tokens on the Airfoil benchmark of LaMO compared with Transolver. Lower relative $l_2$ error indicates better model performance.

| ABLATIONS | | RELATIVE L2 | |
|---|---|---|---|
| | | TRANSOLVER | LaMO (OURS) |
| | 1 | 0.0085 | 0.0079 |
| | 8 | 0.0058 | 0.0050 |
| | 16 | 0.0057 | 0.0048 |
| | 32 | 0.0055 | 0.0045 |
| NUMBER | 64 | 0.0054 | 0.0041 |
| OF LATENT | 96 | 0.0053 | 0.0040 |
| TOKENS | 128 | **0.0052** | **0.0039** |
| | 256 | 0.0058 | 0.0043 |
| | 512 | 0.0059 | 0.0048 |

olutions, depths, and embedding dimensions. As shown in Figure 2(a), LaMO achieves superior performance on the Darcy flow benchmark even with 40% of the training data, demonstrating its remarkable efficiency in low-data regimes compared to second-best baseline models (Wu et al., 2024). This demonstrates LaMO's efficiency in learning meaningful representation with fewer data points, making it ideal for real-world applications with limited data. Additionally, we evaluated LaMO under different benchmark resolutions (Figure 2 (b)), depth (Figure 2 (c)), and embedding dimensions (Figure 2 (d)). The results indicate that the initial LaMO configuration maintains consistent performance across varying resolutions, depths, and embedding dimensions compared with the second-best operator, Transolver. These findings suggest that LaMO exhibits robust scalability properties and can serve as a large-scale pre-trained PDE solver, paving the way for its use as a foundation model in SciML applications. The appendix Subsection E.5 presents more experiments across the benchmark.

***Visual Demonstrations:*** To visualize the performance of LaMO compared to Transolver, we plot heatmaps in Figure 3 for an intuitive performance comparison. It is observed that LaMO significantly outperforms Transolver in capturing medium boundaries in Darcy flow problems (Figure 3(a)). Specifically, LaMO excels at resolving discontinuity boundaries and junctions with fewer artifacts than Transolver (Figure 3(b)). Furthermore, in time-dependent PDEs, such as the Navier-Stokes equations, LaMO demonstrates superior performance in accurately capturing turbulent fluid flow dynamics (Figure 3(c)). It handles boundaries and medium transitions more effectively, showcasing its robustness over Transolver in such scenarios. The appendix Section F presents more heat map plots.

***Performance on Turbulent Fluid Dynamics:*** As shown in Table 1, LaMO demonstrates superior performance on Navier–Stokes (Reynold number $10^5$), a highly turbulent flow, achieving an average gain of $51.9\%$ over the second-best baseline. For lower turbulence (Reynold number $10^4$),

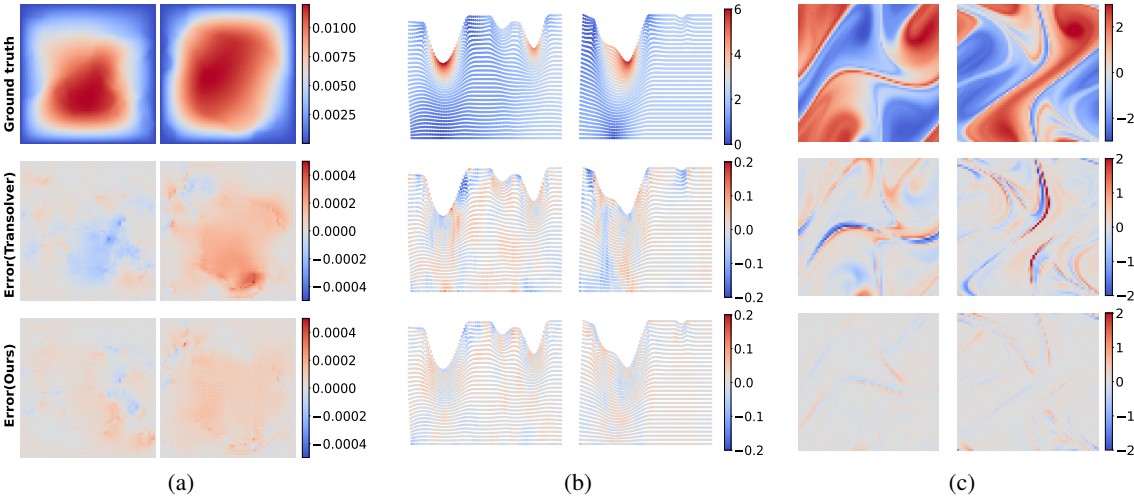

(a)     (b)     (c)

*Figure 3.* Visual Comparison. The (**Top Row**) displays the ground truth, the (**Middle Row**) presents the error heatmap of Transolver, and the (**Bottom Row**) presents the error heatmap for LaMO on (a) Darcy Flow, (b) Plasticity, and (c) Navier-Stokes benchmark.

LaMO achieves a relative $l_2$ error improvement of 75% ($0.0454 \rightarrow 0.0117$) compared to Transolver. Thus validating it as consistent across the turbulent regions.

***Latent Tokens:*** Using latent tokens formed on a regular grid, as in Perceiver, we observed that while bidirectional SSM marginally improved performance $0.0059 \rightarrow 0.0050$, compared to patches ($0.0039$), it lacked scalability with data resolution, with increased computation. It highlights the significance of ViT-type latent tokens in scaling operators to higher resolutions, making them an optimal choice for foundational models, as demonstrated in transformer-based foundation models (Poseidon (Herde et al., 2024)).

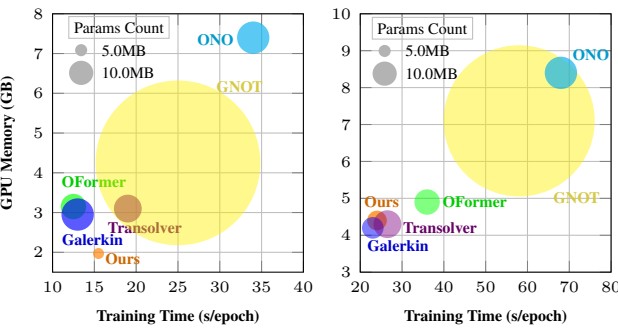

*Figure 4.* Efficiency comparison of top five baselines on (a) Darcy and (b) Airfoil benchmark per epoch, respectively.

***Efficiency:*** To further analyze the performance of the proposed model, we present its efficiency metrics in Figure 4 compared with transformer-based baselines. Specifically, on the Darcy flow and Navier-Stokes benchmark, LaMO utilizes 3× fewer parameters and is 1.8× faster than transformer-based baselines. Compared to other baselines, such as

GNOT and ONO, LaMO exhibits an even more significant improvement, requiring 5–7× fewer parameters and achieving a 3× speedup in running time. LaMO presents favorable efficiency considering the running time, GPU memory consumption, and model parameters compared to Transolver, and it achieves better results. These results highlight the efficiency of LaMO in solving PDEs.

***Interpretability:*** To further interpret the performance, we analyze inter-cosine similarity across layers. As shown in the appendix Subsection E.6, LaMO demonstrates a reduction in inter-cosine similarity across layers compared to Transolver, indicating better representation learning. This improvement can be attributed to the absence of the softmax operation, which in Transolver has been identified as causing over-smoothing issues affecting its ability to capture the intrinsic relationship (Ali et al., 2023; Wang et al., 2022).

## 5. Conclusion and Future Work

In summary, we introduce a new approach to solving PDEs by introducing the SSM-based Neural Operator. The proposed method leverages SSMs to achieve superior performance while ensuring practical efficiency. The theoretical analysis reveals the equivalence of LaMO with kernel integration. Through extensive experiments, we demonstrate that SSM-based operators outperform transformer-based counterparts in performance and computational efficiency. While the current work establishes the potential of SSM-based neural operators, certain aspects remain unexplored. For instance, compatibility with pretraining paradigms using SSM as a backbone has yet to be explored. Future work aims to extend the application of SSM-based operators as foundation models for PDE solutions.

## Acknowledgement

This work was partly supported by the Indian Institute of Science under a start-up grant to set up the GPU computing infrastructure. Prathosh was supported by the Infosys Foundation Young Investigator Award. Anoop acknowledges support from the Alexander von Humboldt Foundation and the Google Research Scholar Award. The Government of India supported Karn through the Prime Minister's Research Fellowship (PMRF).

## Impact Statement

This paper introduces a new neural operator to advance deep learning research in solving PDEs. The method leverages state-space models to capture correlations for long-range dependencies, addressing the limitations of transformer-based operators. Our model demonstrates superior performance in PDE tasks, showing potential for industrial applications and scalability for foundation models. This includes applications for several practical large-scale problems, including, but not limited to, weather forecasting, pollution forecasting, and electronic structure, to name a few. The focus remains on scientific challenges, with no identified ethical risks in this work. Thus, we believe there are no potential ethical risks to our work in society.

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

## A. Table of Notations

Table 4 provides a comprehensive list of notations used throughout the main content for clarity and reference.

*Table 4.* The following table summarizes the notations and symbols used throughout the main text for consistency and clarity.

| NOTATIONS | DESCRIPTIONS |
|---|---|
| LaMO | Latent Mamba Operator |
| PDEs | Partial Differential Equations |
| ODEs | Ordinary Differential Equations |
| SSMs | State Space Models |
| ZOH | Zero Order Hold |
| Linear | Feedforward Neural Network |
| $D \subset \mathbb{R}^d$ | Spatial Domain for the PDE |
| $x \in D$ | Spatial Domain Points |
| $a \in A = (D; \mathbb{R}^{d_a})$ | Input Functions Coefficient |
| $d_a$ | Dimension of Input Function $a(x)$ |
| $u \in U = (D; \mathbb{R}^{d_u})$ | Target Functions Solution |
| $d_u$ | Dimension of Output Function $u(x)$ |
| $D_j$ | The Discretization of $(a_j, u_j)$ |
| $\mathcal{G}^\dagger : A \to U$ | Solution Operator |
| $\mathcal{G}_\theta$ | Neural Operator |
| $\mu$ | A Probability Measure on $A$ |
| $\mathcal{L}$ | Loss Function |
| $x(t) \in \mathbb{R}^H$ | System Input Sequence |
| $h(t) \in \mathbb{R}^N$ | System State |
| $y(t) \in \mathbb{R}^M$ | System Output Sequence |
| $x[k] \in \mathbb{R}^H$ | Discrete Input Sequence |
| $h[k] \in \mathbb{R}^N$ | Discrete State |
| $y[k] \in \mathbb{R}^M$ | Discrete Output Sequence |
| $\mathbf{A} \in \mathbb{R}^{N \times N}$ | System Matrix in Continuous SSM |
| $\mathbf{B} \in \mathbb{R}^{N \times H}$ | Input Matrix in Continuous SSM |
| $\mathbf{C} \in \mathbb{R}^{M \times N}$ | Output Matrix in Continuous SSM |
| $\mathbf{D} \in \mathbb{R}^{M \times H}$ | Direct Transition Matrix in Continuous SSM |
| $\bar{\mathbf{A}} \in \mathbb{R}^{N \times N}$ | System Matrix in Discrete SSM |
| $\bar{\mathbf{B}} \in \mathbb{R}^{N \times H}$ | Input Matrix in Discrete SSM |
| $\bar{\mathbf{C}} \in \mathbb{R}^{M \times N}$ | Output Matrix in Discrete SSM |
| $\bar{\mathbf{D}} \in \mathbb{R}^{M \times H}$ | Direct Transition Matrix in Discrete SSM |
| $\Delta \in \mathbb{R}^+$ | Discrete Time Step in Discrete SSM |
| $\mathbf{K}$ | State Kernel in Convolutional SSM |
| $\kappa$ | Kernel Integral Operator |
| $\mathbf{X}$ | Physical Tokens |
| $\mathbf{Z}$ | Latent Tokens |
| $N$ | Number of Training Samples |

## B. Theoretical Insights

This section delves into a rigorous mathematical analysis of the state-space model (SSM) from the perspective of a neural operator. We provide proof of its core properties and present new insights. This comprehensive examination will highlight the theoretical foundations and unique advantages of leveraging SSMs in neural operators. We abuse the notations when there is no misleading context.

**Lemma B.1.** *(Williams et al., 2007) For any differential equation of the following form*

$$h'(t) = Ah(t) + Bx(t), \tag{22}$$

*Its general solution is given as follows:*

$$h(t) = e^{A(t-t_0)}h(t_0) + \int_{t_0}^{t} e^{A(t-s)}Bx(s)\,ds. \tag{23}$$

*Proof:* Consider the $n$-dimensional dynamical system represented by the equation as follows:

$$h'(t) = Ah(t) + Bx(t), \tag{24}$$

where $h(t) \in \mathbb{R}^n$ is the state vector, $A \in \mathbb{R}^{n \times n}$ is the system matrix, $B \in \mathbb{R}^{n \times m}$ is the input matrix, and $x(t) \in \mathbb{R}^m$ represents the input. Rearranging the terms, we get the following:

$$h'(t) - Ah(t) = Bx(t). \tag{25}$$

By multiplying both sides by the integrating factor $e^{-tA}$, we obtain the following:

$$e^{-tA}h'(t) - e^{-tA}Ah(t) = e^{-tA}Bx(t). \tag{26}$$

$$\frac{d}{dt}\left(e^{-tA}h(t)\right) = e^{-tA}Bx(t). \tag{27}$$

Integrating both sides with respect to time over the interval $[0, t]$, we have the following:

$$\int_0^t \frac{d}{ds}\left(e^{-sA}h(s)\right)ds = \int_0^t e^{-sA}Bx(s)ds. \tag{28}$$

$$e^{-tA}h(t) - h(0) = \int_0^t e^{-sA}Bx(s)ds. \tag{29}$$

$$h(t) = e^{tA}h(0) + \int_0^t e^{A(t-s)}Bx(s)ds. \tag{30}$$

This can be rewritten in convolutional form as follows:

$$h(t) = e^{tA}h(0) + (u * x)(t), \tag{31}$$

where $u(t) = Be^{At}$ represents the impulse response function.

Thus, the above equation provides the analytical solution to the $n$-dimensional state-space equation. $\qquad\square$

*Remark* B.2. It is important to note that the solution consists of two components:

- **Initial Response:** $e^{tA}h(0)$, which depends on the initial state.

- **Forced Response:** $\int_0^t e^{A(t-s)}Bx(s)ds$, driven by the input $x(t)$.

Both components involve the exponential matrix function, which encapsulates the system's dynamics, and $Be^{tA}$ denotes the kernel of the SSM.

**Assumption B.3.** We discretize the input signal to incorporate the State Space Model (SSM) into a neural network to obtain a discrete parameter representation equivalent to the continuous SSM. Specifically, the signal $u(t)$ is sampled at uniform intervals of $\Delta$, such that the time $t$ is represented as $k\Delta$, where $k = 0, 1, \ldots$ is a non-negative integer. Utilizing the Zero-Order Hold (ZOH) method, we assume that the signal remains constant within each sampling interval, i.e., $u(t) = u(k\Delta)$ for $t \in [k\Delta, (k+1)\Delta]$.

**Proposition B.4.** *The Zero-Order Hold (ZOH) discretization of continuous-time system parameters can be represented as:*

$$\begin{aligned}\overline{\mathbf{A}} &= e^{\Delta\mathbf{A}}, \\ \overline{\mathbf{B}} &= (\Delta\mathbf{A})^{-1}\left(e^{\Delta\mathbf{A}} - \mathbf{I}\right)\mathbf{B},\end{aligned} \tag{32}$$

*where $\Delta$ denotes the discretization time step, $\mathbf{I}$ is the identity matrix, and $\mathbf{A}, \mathbf{B}$ are the continuous-time system matrices.*

*Proof:* From the previous proposition, the general equation of the continuous state-space model (SSM) is given by:

$$h(t) = e^{A(t)}h(t_0) + \int_{t_0}^{t} e^{A(t-s)}Bx(s)\,ds. \tag{33}$$

Assuming a zero-order hold (ZOH) on the input $x(s)$ between $t \in [k\Delta, (k+1)\Delta]$, the system evolves as:

$$h((k+1)\Delta) = e^{A\Delta}h(k\Delta) + \int_{k\Delta}^{(k+1)\Delta} e^{A((k+1)\Delta - s)}Bx(k\Delta)\,ds. \tag{34}$$

By defining $h((k+1)\Delta) = h_{k+1}$, $h(k\Delta) = h_k$, and $x(k\Delta) = x_k$, the equation simplifies to following:

$$h_{k+1} = e^{A\Delta}h_k + \left( \int_{k\Delta}^{(k+1)\Delta} e^{A((k+1)\Delta - s)}B\,ds \right) x_k. \tag{35}$$

This corresponds to the standard discrete state-space model with parameters as follows:

$$\overline{A} = e^{A\Delta}, \quad \overline{B} = \int_{k\Delta}^{(k+1)\Delta} e^{A((k+1)\Delta - s)}B\,ds. \tag{36}$$

Now, assuming $A$ is invertible, we can simplify $\overline{B}$ as follows:

$$\overline{B} = \int_{0}^{\Delta} e^{As}B\,ds. \tag{37}$$

$$\overline{B} = \int_{0}^{\Delta} A^{-1}\frac{d}{ds}(e^{As})\,ds\,B. \tag{38}$$

$$\overline{B} = A^{-1}(e^{A\Delta} - I)B. \tag{39}$$

Thus, under the assumption of $A$ being invertible, the continuous SSM can be discretized with the parameters:

$$\overline{A} = e^{A\Delta}, \quad \overline{B} = A^{-1}(e^{A\Delta} - I)B. \tag{40}$$

$\square$

*Remark B.5.* Further $\overline{B} = A^{-1}(e^{A\Delta} - I)B$ can be further simplified by truncating higher order terms as follows:

$$\overline{B} = A^{-1}(e^{A\Delta} - I)B = A^{-1}(I + A\Delta + \mathcal{O}(\Delta^2) - I)B = \Delta B. \tag{41}$$

**Corollary B.6.** *The discretization method described above aligns with the Euler method, which approximates the matrix exponential by truncating its Taylor series expansion to the first-order term.*

*Proof:* By substituting the discrete parameters into the SSM equation, we derive the following:

$$h(t + \Delta t) = \overline{A}h(t) + \overline{B}x(t), \tag{42}$$

$$= e^{A\Delta t}h(t) + \Delta t Bx(t), \tag{43}$$

$$= \big(I + \Delta t A + \mathcal{O}(\Delta t^2)\big)h(t) + \Delta t Bx(t), \tag{44}$$

$$= h(t) + \Delta t Ah(t) + \Delta t Bx(t), \tag{45}$$

$$= h(t) + \Delta t\big(Ah(t) + Bx(t)\big), \tag{46}$$

$$= h(t) + \Delta t h'(t). \tag{47}$$

The above illustrates that the update step corresponds to the Euler method's first-order approximation of the dynamics. $\square$

*Remark B.7.* The above corollary establishes the equivalence between the Euler and Zero-Order Hold (ZOH) methods in solving PDEs. While the Euler method truncates higher-order terms from the Taylor series, the ZOH method retains them, making it a more generalized and accurate approach for solving PDEs. This distinction is crucial for understanding SSMs' performance from the Euler method's perspective.

**Lemma B.8.** *(Williams et al., 2007) Any continuous nonlinear differentiable dynamical system described by*

$$\dot{h}(t) = f(h(t), x(t), t), \tag{48}$$

*can be approximated by its linear state-space model (SSM) representation*

$$\dot{h}(t) = Ah(t) + Bx(t) + O(h, x), \tag{49}$$

*where the Jacobian matrices $A$ and $B$ are given by:*

$$A = \frac{\partial f}{\partial h}(\tilde{h}(t), \tilde{x}(t), t), \quad B = \frac{\partial f}{\partial x}(\tilde{h}(t), \tilde{x}(t), t), \tag{50}$$

*and $O(h, x)$ represents higher-order infinitesimal terms.*

**Proof:** Using the Taylor series expansion around $\left(\tilde{h}(t), \tilde{x}(t)\right)$, we expand $f(h(t), x(t), t)$ as follows:

$$f(h(t), x(t), t) = f(\tilde{h}(t), \tilde{x}(t), t) + \frac{\partial f}{\partial h}(\tilde{h}(t), \tilde{x}(t), t)\big(h(t) - \tilde{h}(t)\big) + \frac{\partial f}{\partial x}(\tilde{h}(t), \tilde{x}(t), t)\big(x(t) - \tilde{x}(t)\big) + \text{HOT}, \tag{51}$$

where HOT denotes higher-order terms.

By defining the Jacobian matrices as follows:

$$A(t) = \frac{\partial f}{\partial h}(\tilde{h}(t), \tilde{x}(t), t), \quad B(t) = \frac{\partial f}{\partial x}(\tilde{h}(t), \tilde{x}(t), t), \tag{52}$$

and rearranging terms, we obtain the following:

$$\dot{h}(t) = Ah(t) + Bx(t) + O(h, x). \tag{53}$$

Thus, a linear SSM with higher-order corrections can approximate continuous nonlinear differentiable dynamics. $\square$

*Remark* B.9. (Alonso et al., 2024) A nonlinear continuous differential system with the above dynamics has long-range memory, i.e., it captures information from past inputs if all eigenvalues of $A$ are within the unit circle. Mathematically, it should satisfy the following conditions:

$$|\text{eig}(A)| \leq 1 \quad \text{and} \quad |\text{eig}(A)| \approx 1 \quad \forall\, \text{eig}(A).$$

**Lemma B.10.** *The SSM operator is a Monte Carlo approximation of an integral operator.*

**Proof:** Let $(\Omega, \mathcal{F}, \mu)$ be a probability space, where $\Omega$ is a measurable space equipped with a probability measure $\mu$. Let $u : \Omega \to \mathbb{R}^C$ be a function in the Hilbert space $\mathcal{L}^2(\Omega, \mu; \mathbb{R}^C)$. Let define, the integral operator $\mathcal{G} : \mathcal{L}^2(\Omega, \mu; \mathbb{R}^C) \to \mathcal{L}^2(\Omega, \mu; \mathbb{R}^C)$ as follows:

$$\mathcal{G}(u)(x) = \int_{\Omega} \kappa(x, y) u(y)\, \mu(dy), \tag{54}$$

where $\kappa : \Omega \times \Omega \to \mathbb{R}$ is a measurable kernel function.

Now let's discretize the domain, i.e., consider a partition of $\Omega$ into $N$ distinct mesh points $\{y_i\}_{i=1}^N$ in particular order sequence, and approximate the measure $\mu$ by a uniform measure over these points. The integral operator $\mathcal{G}$ can then be approximated via Monte Carlo integration as follows:

$$\mathcal{G}(u)(x) \approx \frac{|\Omega|}{N} \sum_{i=1}^{N} \kappa(x, y_i) u(y_i), \tag{55}$$

where $|\Omega| = \int_{\Omega} 1\, d\mu$ denotes the total measure of $\Omega$.

In SSM, the integral kernel $\kappa$ is parameterized as follows:

$$\kappa(y_i, y_j) = \mathbf{W}_C(y_i) \underbrace{\left( \prod_{y_k \leq y_i} \overline{A} \right)}_{\text{Relative Position}} \mathbf{W}_{\overline{B}}(y_j) \tag{56}$$

the relative position denotes the matrix exponential of $\overline{A}$ depends on the position, $y_k \leq y_i$ denotes the relative distance between the tokens and $\mathbf{W}_C$ and $\mathbf{W}_{\overline{B}}$ denotes the SSM input dependent weights.

Substituting this parameterization into the Monte Carlo approximation, we obtain the following:

$$\mathcal{G}(u)(x) \approx \frac{|\Omega|}{N} \sum_{i=1}^{N} \mathbf{W}_C(y_i) \left( \prod_{y_k \leq x} \overline{A} \right) \mathbf{W}_{\overline{B}}(x) u(y_i). \tag{57}$$

This expression demonstrates that the SSM operator can be interpreted as a Monte Carlo approximation of the integral operator $\mathcal{G}$, where the state-space dynamics parameterize the kernel $\kappa$. □

**Lemma B.11.** *(Wu et al., 2024) If $\Omega$ is a countable domain, the latent domain $\Omega_s$ is isomorphic to $\Omega$.*

**Proof:** Let $x_i \in \Omega$ denote the $i$-th element in the input domain, and let $z_j$ represent the $j$-th latent token in the latent domain $\Omega_s$. The latent weight of $x_i$ with respect to $z_j$ is denoted by $w_{x_i, z_j} \in \mathbb{R}$. Given a constant $K \geq 1$ and $K \in \mathbb{N}$, we define a projection $f : \Omega \to \Omega_l$ as follows:

$$f(x_i) = \arg\max_{z_j} w_{x_i, z_j}, \tag{58}$$

where the mapping is constrained such that:

$$\lfloor (i-1)/K \rfloor \cdot K < j \leq (\lfloor (i-1)/K \rfloor + 1) \cdot K, \tag{59}$$

and the $j$-th latent token $z_j$ has not been assigned a projection previously.

This construction guarantees a bijective mapping between elements of the input domain $\Omega$ and latent token in the latent domain $\Omega_s$, ensuring that the cardinalities of the two domains are equivalent. Hence, $\Omega$ is isomorphic to $\Omega_s$, i.e., $\Omega \sim \Omega_s$.

Using the above results, we will show that SSM is an integral kernel on $\Omega$ similar to transformer-based operator (Wu et al., 2024; Cao, 2021; Wu et al., 2023).

**Theorem B.12 (SSM as an equivalent integral kernel on $\Omega$).** *Let $\Omega \subseteq \mathbb{R}^n$ be a bounded domain, $\boldsymbol{a} : \Omega \to \mathbb{R}^d$ be a given input function and $\mathbf{x} \in \Omega$ be a mesh point. An SSM (Structured State-Space Model) layer approximates the linear integral operator $\mathcal{G} : L^2(\Omega, \mathbb{R}^d) \to L^2(\Omega, \mathbb{R}^d)$, defined as follows:*

$$\mathcal{G}(\boldsymbol{a})(\mathbf{x}) = \int_\Omega \kappa(\mathbf{x}, y)\, \boldsymbol{a}(y)\, \mathrm{d}y, \tag{60}$$

*where $\kappa : \Omega \times \Omega \to \mathbb{R}^{d \times d}$ is the kernel function characterizing the operator $\mathcal{G}$.*

**Proof:** According to Lemma A.3, we can obtain an isomorphic projection $f$ between a countable input domain $\Omega$ and the latent domain $\Omega_z$. Suppose that the latent weight $w_{*,*} : \Omega \times \Omega_z \to \mathbb{R}$ is smooth in both $\Omega$ and $\Omega_z$, where $\Omega$ and $\Omega_z$ denote the continuation of $\Omega$ and $\Omega_z$, respectively. Then, we can obtain $f$ as a diffeomorphism projection.

Then, we define the value function $u_s$ on the latent token domain $\Omega_z$ as follows:

$$u(z) = \int_\Omega w_{x,z} u(x)\, dx \tag{61}$$

which corresponds to the latent token definition in Eq. (2). Based on the above assumptions and definitions, we have:

$$G(u)(x) = \int_\Omega \kappa(x, y) u(y) \, dy \tag{62}$$

$$= \int_{\Omega_z} \kappa_{\text{mz}}(x, y_z) u_z(y_z) \, df^{-1}(y_z) \qquad (\kappa_{\text{mz}}(\cdot, \cdot) : \Omega \times \Omega_z \to \mathbb{R}^{C \times C} \text{ is a kernel function}) \tag{63}$$

$$= \int_{\Omega_z} \kappa_{\text{mz}}(x, y_z) u_z(y_z) \, |\det(\nabla_{y_z} f^{-1}(y_z))| d(y_z) \tag{64}$$

$$= \int_{\Omega_z} \left( \int_{\Omega_z} w_{x, y_s'} \kappa_{\text{zz}}(y_z', y_s) dy_z' \right) u_z(y_z) \, |\det(\nabla_{y_z} f^{-1}(y_z))| d(y_z) \tag{65}$$

$$= \int_{\Omega_z} \left( \int_{\Omega_z} w_{x, y_z'} \kappa_{\text{zz}}(y_z', y_z) dy_z' \right) u_z(y_z) \, |\det(\nabla_{y_z} f^{-1}(y_z))| d(y_z) \tag{66}$$

$$= \int_{\Omega_z} \underbrace{w_{x, y_z'}}_{\text{Decoder}} \int_{\Omega_s} \underbrace{\kappa_{\text{zz}}(y_z', y_z)}_{\text{SSM among the latent tokens}} \underbrace{u_z(y_z)}_{\text{Latent token}} |\det(\nabla_{y_z} f^{-1}(y_z))| d(y_z) dy_z' \qquad (\text{Lemma B.10}) \tag{67}$$

$$\approx \underbrace{\sum_{j=1}^M w_{i,j}}_{\text{Eq. (15)}} \underbrace{\sum_{t=1}^M \left( (\mathbf{W_C} u_z(z_{z,j})) \left( \prod_{k : z_{z,j} \leq z_{z,t}} \text{Diag}(\overline{A}) \right) (\mathbf{W_B} u_z(z_{z,t})) \right)}_{\text{Eq. (13)}} \underbrace{\left( \sum_{p=1}^N w_{p,t} u(\mathbf{x}_p) \right)}_{\text{Eq. (10)}} \tag{68}$$

$$= \sum_{j=1}^M w_{i,j} \sum_{t=1}^M \left( \mathbf{C}_j \left( \prod_{j \leq t} \text{Diag}(\overline{A}) \right) (\mathbf{B}_t) \right) \mathbf{z}_t, \tag{69}$$

where $\kappa_{\text{mz}}$ denotes the kernel function between mesh points and latent tokens, and $\kappa_{\text{zz}}$ is the kernel defined among latent tokens. For simplicity, we take $|\det(\nabla_{y_z} f^{-1}(y_z))| = 1$ for simplification. Different from the kernel integral among mesh points, the usage of Lemma B.10 here is based on the Monte-Carlo approximation in the latent domain. $\qquad \square$

*Remark* B.13. The kernel function $\kappa(x, y)$ encapsulates the interactions between the evaluation point $x$ and the domain points $y \in \Omega$. The SSM layer implements a numerical approximation of the integral operator using a finite-dimensional representation of the input function $a$ over the discretized domain $\Omega$. Specifically, the SSM architecture constructs a learned representation of $\kappa$ and combines it with the discretized function $a$ to approximate the integral.

In the subsequent section, we comprehensively analyze the learning mechanism underlying the State-Space Model (SSM) kernel and explore its theoretical and empirical connections to transformer-based architectures.

**Matrices in Structures State Space models:** Given an input sequence $X := [x_1, \cdots, x_L] \in \mathbb{R}^{L \times D}$ consisting of $L$ feature vectors, the hidden state of each vector is denoted as follows:

$$h_1 = \overline{B}_1 x_1, \tag{70}$$

$$h_2 = \overline{A}_2 h_1 + \overline{B}_2 x_2 = \overline{A}_2 \overline{B}_1 x_1 + \overline{B}_2 x_2, \tag{71}$$

$$h_3 = \overline{A}_3 h_2 + \overline{B}_3 x_3 = \overline{A}_3 \overline{A}_2 \overline{B}_1 x_1 + \overline{A}_3 \overline{B}_2 x_2 + \overline{B}_3 x_3, \tag{72}$$

$$\cdots$$

$$h_L = \overline{A}_L h_{L-1} + \overline{B}_L x_L = \overline{A}_L \overline{A}_{L-1} \cdots \overline{A}_2 \overline{B}_1 x_1 + \overline{A}_L \overline{A}_{L-1} \cdots \overline{A}_3 \overline{B}_2 x_2 + \cdots + \overline{B}_L x_L. \tag{73}$$

The above recurrence can be rewritten in matrix form as:

$$H = [h_1, h_2, h_3, \cdots, h_L]^\top = \begin{bmatrix} \overline{B}_1 & 0 & 0 & \cdots & 0 \\ \overline{A}_2 \overline{B}_1 & \overline{B}_2 & 0 & \cdots & 0 \\ \overline{A}_3 \overline{A}_2 \overline{B}_1 & \overline{A}_3 \overline{B}_2 & \overline{B}_3 & \cdots & 0 \\ \vdots & \vdots & \vdots & \ddots & \vdots \\ \prod_{j=2}^L \overline{A}_j \overline{B}_1 & \prod_{j=3}^L \overline{A}_j \overline{B}_2 & \prod_{j=4}^L \overline{A}_j \overline{B}_3 & \cdots & \overline{B}_L \end{bmatrix} \begin{bmatrix} x_1 \\ x_2 \\ x_3 \\ \vdots \\ x_L \end{bmatrix}. \tag{74}$$

For the output sequence $Y := [y_1, \cdots, y_L]^\top$, each vector $y_i$ ($i = 1, \cdots, L$) is expressed as follows:

$$y_i = C_i h_i, \tag{75}$$

where and in matrix form, the above equation can be compactly rewritten as follows:

$$Y = \begin{bmatrix} C_1 & 0 & 0 & \cdots & 0 \\ 0 & C_2 & 0 & \cdots & 0 \\ 0 & 0 & C_3 & \cdots & 0 \\ \vdots & \vdots & \vdots & \ddots & \vdots \\ 0 & 0 & 0 & \cdots & C_L \end{bmatrix} \begin{bmatrix} h_1 \\ h_2 \\ h_3 \\ \vdots \\ h_L \end{bmatrix} = CH. \tag{76}$$

Substituting $H$ from the Eq, we obtain the following relation:

$$Y = C \begin{bmatrix} \overline{B}_1 & 0 & 0 & \cdots & 0 \\ \overline{A}_2\overline{B}_1 & \overline{B}_2 & 0 & \cdots & 0 \\ \overline{A}_3\overline{A}_2\overline{B}_1 & \overline{A}_3\overline{B}_2 & \overline{B}_3 & \cdots & 0 \\ \vdots & \vdots & \vdots & \ddots & \vdots \\ \prod_{j=2}^{L} \overline{A}_j\overline{B}_1 & \prod_{j=3}^{L} \overline{A}_j\overline{B}_2 & \prod_{j=4}^{L} \overline{A}_j\overline{B}_3 & \cdots & \overline{B}_L \end{bmatrix} \begin{bmatrix} x_1 \\ x_2 \\ x_3 \\ \vdots \\ x_L \end{bmatrix}. \tag{77}$$

In general, $t = 1, \cdots, L$ we have following recurrence for hidden states and output,

$$h_t = \sum_{j=1}^{t} \prod_{k=j+1}^{t} \overline{A}_k \overline{B}_j x_j \tag{78}$$

$$y_t = C_t \sum_{j=1}^{t} \prod_{k=j+1}^{t} \overline{A}_k \overline{B}_j x_j. \tag{79}$$

which can be further simplified and written as follows:

$$Y = \begin{bmatrix} C_1\overline{B}_1 & 0 & 0 & \cdots & 0 \\ C_2\overline{A}_2\overline{B}_1 & C_2\overline{B}_2 & 0 & \cdots & 0 \\ C_3\overline{A}_3\overline{A}_2\overline{B}_1 & C_3\overline{A}_3\overline{B}_2 & C_3\overline{B}_3 & \cdots & 0 \\ \vdots & \vdots & \vdots & \ddots & \vdots \\ C_L \prod_{j=2}^{L} \overline{A}_j\overline{B}_1 & C_L \prod_{j=3}^{L} \overline{A}_j\overline{B}_2 & C_L \prod_{j=4}^{L} \overline{A}_j\overline{B}_3 & \cdots & C_L\overline{B}_L \end{bmatrix} \begin{bmatrix} x_1 \\ x_2 \\ x_3 \\ \vdots \\ x_L \end{bmatrix}. \tag{80}$$

This can be compactly written as:

$$Y = MX = SSM(\overline{A}, \overline{B}, C) \tag{81}$$

where $M \in \mathbb{R}^{L \times L}$ represents the matrix. Hence, the $M$ can be seen as a controlled linear operator. The above Equation can be seen as a variant of self-attention, specifically, the casual version of attention. The element of $M$ are as follows:

$$M_{i,j} = C_i \prod_{k=j+1}^{i} \overline{A}_k \overline{B}_j \tag{82}$$

*Remark* B.14. Linear SSM is linear shift-invariant and thus is a causal system.

**Similarity with Attention:** The input sequence $X \in \mathbb{R}^{L \times D}$ are linearly projected into Query $\mathbf{Q} \in \mathbb{R}^{L \times D}$, Key $\mathbf{K} \in \mathbb{R}^{L \times D}$, and Value $\mathbf{V} \in \mathbb{R}^{L \times D}$.

$$Q = XW_Q^\top, \quad K = XW_K^\top, \quad V = XW_V^\top, \tag{83}$$

with $W_Q, W_K \in R^{D \times D}$ and $W_V \in R^{D \times D}$ as the corresponding weight matrices. More specifically, $\mathbf{Q} := [\mathbf{q}_1, \ldots, \mathbf{q}_L]^\top, \mathbf{K} := [\mathbf{k}_1, \ldots, \mathbf{k}_L]^\top, \mathbf{V} := [\mathbf{v}_1, \ldots, \mathbf{v}_L]^\top$, where $\mathbf{q}_i, \mathbf{k}_i, \mathbf{v}_i$ (for $i = 1, \ldots, L$) represent the query, key, and

value vectors, respectively, for input $x_i$. Based on $\mathbf{Q}$ and $\mathbf{K}$, the attention matrix $\mathbf{S} \in \mathbb{R}^{L \times L}$ contains the correlations between all query and key vectors, with a softmax function applied to each row of $\mathbf{S}$ which reduce it to transition matrix (row stochastic matrix):

$$\mathbf{S} = \text{softmax}\left(\frac{\mathbf{QK}^\top}{\sqrt{D}}\right). \tag{84}$$

where the above matrix can be written as follows (for clarity, we have ignored the normalization constant $\sqrt{(d)}$):

$$\mathbf{S} = \begin{bmatrix} \frac{\exp(\mathbf{q}_1 \cdot \mathbf{k}_1)}{\sum\limits_{j=1}^{L} \exp(\mathbf{q}_1 \cdot \mathbf{k}_j)} & \frac{\exp(\mathbf{q}_1 \cdot \mathbf{k}_2)}{\sum\limits_{j=1}^{L} \exp(\mathbf{q}_1 \cdot \mathbf{k}_j)} & \cdots & \frac{\exp(\mathbf{q}_1 \cdot \mathbf{k}_L)}{\sum\limits_{j=1}^{L} \exp(\mathbf{q}_1 \cdot \mathbf{k}_j)} \\ \vdots & \vdots & \ddots & \vdots \\ \frac{\exp(\mathbf{q}_L \cdot \mathbf{k}_1)}{\sum\limits_{j=1}^{L} \exp(\mathbf{q}_L \cdot \mathbf{k}_j)} & \frac{\exp(\mathbf{q}_L \cdot \mathbf{k}_2)}{\sum\limits_{j=1}^{L} \exp(\mathbf{q}_L \cdot \mathbf{k}_j)} & \cdots & \frac{\exp(\mathbf{q}_L \cdot \mathbf{k}_L)}{\sum\limits_{j=1}^{L} \exp(\mathbf{q}_L \cdot \mathbf{k}_j)} \end{bmatrix}. \tag{85}$$

Each element $S_{i,j}$ (for $i, j = 1, \ldots, L$) represents the attention score between $\mathbf{q}_i$ and $\mathbf{k}_j$.

For the output sequence $Y := [y_1, \cdots, y_L]^\top$, is calculated based on Attention Matrix ($\mathbf{S}$) as follows:

$$\mathbf{Y} = \mathbf{SV} \tag{86}$$

which can be rewritten in term of $\mathbf{X}$ as follows:

$$\mathbf{Y} = \mathbf{SXW^T} \tag{87}$$

It follows that each output vector $\mathbf{y}_i$ (for $i = 1, \ldots, L$) can be written in vector form as:

$$\mathbf{y}_i = \sum_{j=1}^{L} S_{i,j} \mathbf{v}_j \tag{88}$$

which implies the output $y_i$ is linear combination of vectors $v_j (j = 1, ..., L)$ with coefficient $S_{i,j}$. The greater the attention score, the larger its influence on the output.

*Remark* B.15. From both the matrix form of SSM and attention, we observe that both of them satisfy the pseudo-linear form, i.e as follows:

$$Y = M(X)X \tag{89}$$

we can observe that both are linear equations, but the weight matrix is data-dependent.

**Interpreting the Hidden Matrices of SSM:** From Equation 82 of SSM, we have the element of the matrix as follows:

$$M_{i,j} = C_i \underbrace{\prod_{k=j+1}^{i} \overline{A}_k}_{\text{Discount Factor}} \overline{B}_j \tag{90}$$

Let abbreviate $H_{i,j} = \prod_{k=j+1}^{i} \overline{A}_k$ above equation can be written as,

$$M_{i,j} = C_i H_{i,j} B_j \tag{91}$$

In Structured State-Space Models (SSMs), the softmax normalization typically used in transformers is removed. Additionally, SSMs incorporate a masking mechanism that introduces *input-dependent relative positional encodings* (denoted as $H_{i,j}$), in contrast to transformers, where positional encodings are typically fixed. This masking mechanism can be interpreted as a "discount factor" that modulates interactions based on the relative distance between positions $i$ and $j$, formulated as:

$$H_{i,j} = \prod_{k=j+1}^{i} \overline{A}_k. \tag{92}$$

which in the context of attention mechanisms, this input-dependent positional mask plays a crucial role in encoding the "selectivity" of SSMs in neural operator (data-dependent kernel), thereby influencing their ability to capture dependencies in modeling in PDEs solution (Katsch, 2023; Dao & Gu, 2024).

## C. Details of Benchmark

This section provides a detailed overview of the benchmark datasets and the specific tasks associated with each benchmark.

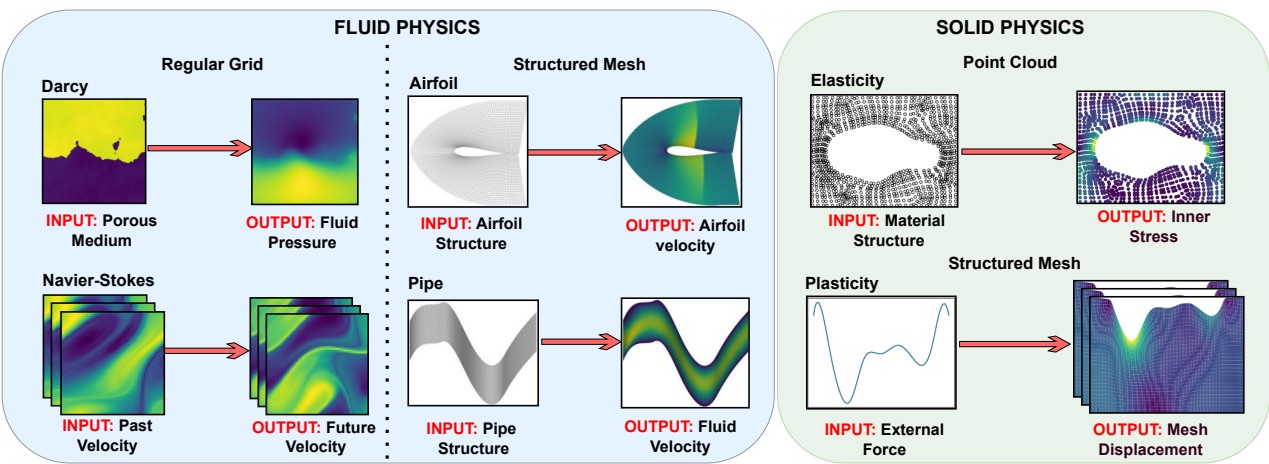

*Figure 5.* The diagram presents an overview of the Neural Operator learning task benchmark PDEs classified as Fluid Physics and Solid Physics. **(Top Row)** showcases three specific PDEs: Darcy Flow, Airfoil, and Elasticity. **(Bottom Row)**, an additional set of three PDEs is shown: Navier Stokes, Pipe, and Plasticity.

*Table 5.* The benchmark details follow the settings from (Li et al., 2022a), with input-output resolutions presented as (temporal, spatial, variate). The "/" denotes the absence of that dimension.

| | OVERVIEW | SOLID PHYSICS | | FLUID PHYSICS | | | |
| | | ELASTICITY | PLASTICITY | NAVIER–STOKES | AIRFOIL | PIPE | DARCY |
|---|---|---|---|---|---|---|---|
| **PHYSICS** | TASK | ESTIMATE STRESS | MODEL DEFORMATION | MODEL VISCOUS FLOW | ESTIMATE VELOCITY | | ESTIMATE PRESSURE |
| | INPUT | MATERIAL STRUCTURE | EXTERNAL FORCE | PAST VELOCITY | STRUCTURE | | POROUS MEDIUM |
| | OUTPUT | INNER STRESS | MESH DISPLACEMENT | FUTURE VELOCITY | MACH NUMBER | FLUID VELOCITY | FLUID PRESSURE |
| **DATA** | GEOMETRY | POINT CLOUD | STRUCTURED MESH | REGULAR GRID | STRUCTURED MESH | | REGULAR GRID |
| | DIMENSION | 2D | 2D + TIME | | 2D | | |
| | TRAIN SET SIZE | 1000 | 900 | 1000 | 1000 | 1000 | 1000 |
| | TEST SET SIZE | 200 | 80 | 200 | 100 | 200 | 200 |
| | INPUT TENSOR | $(/, 972, 2)$ | $(/, 101 \times 31, 2)$ | $(10, 64 \times 64, 1)$ | $(/, 221 \times 51, 2)$ | $(/, 129 \times 129, 2)$ | $(/, 85 \times 85, 1)$ |
| | OUTPUT TENSOR | $(/, 972, 1)$ | $(20, 101 \times 31, 4)$ | $(10, 64 \times 64, 1)$ | $(/, 221 \times 51, 1)$ | $(/, 129 \times 129, 1)$ | $(/, 85 \times 85, 1)$ |

Table 5 and Fig. 5 provide a comprehensive overview of the benchmark details. The categorization of the generation details is based on the governing PDEs, which are as follows:

**Elasticity** (Li et al., 2022a): This benchmark aims to estimate the internal stress of an elastic material based on its structure, discretized into 972 points. For each sample, the input is a tensor with a shape of $972 \times 2$, representing the 2D position of each discretized point. The output is the stress at each point, formatted as a tensor of shape $972 \times 1$. In the experiment, 1000 samples with varying structures are generated for training, while 200 samples are used for testing.

**Plasticity** (Li et al., 2022a): This benchmark focuses on predicting the future deformation of a plastic material impacted from above by a die with an arbitrary shape. For each sample, the input is the die's shape, discretized into a structured mesh and represented as a tensor with a shape of $101 \times 31$. The output is the deformation of each mesh point over 20 future timesteps, recorded as a tensor of shape $20 \times 101 \times 31 \times 4$, capturing deformation in four directions. In the experiment, 900 samples with varying die shapes are used for training, while 80 additional samples are reserved for testing.

**Navier-Stokes** (Li et al., 2020): 2D Navier-Stokes equation mathematically describes the flow of a viscous, incompressible

fluid in vorticity form on the unit torus as follows:

$$\partial_t w(x,t) + u(x,t) \cdot \nabla w(x,t) = \nu \Delta w(x,t) + f(x), \quad x \in (0,1)^2, \, t \in (0,T] \tag{93}$$

$$\nabla \cdot u(x,t) = 0, \quad x \in (0,1)^2, \, t \in [0,T] \tag{94}$$

$$w(x,0) = w_0(x), \quad x \in (0,1)^2 \tag{95}$$

where, $u$ represents the velocity field, $w = \nabla \times u$ is the vorticity, $w_0$ is the initial vorticity, $\nu$ is the viscosity coefficient, and $f$ is the forcing function. In this dataset, the viscosity ($\nu$) is fixed at $10^{-5}$, and the 2D field has a resolution of $64 \times 64$. Each sample within the dataset comprises 20 consecutive frames. The objective is to predict the subsequent ten frames based on the preceding ten. The experiment uses 1000 fluid samples with different initial conditions for training, while 200 additional samples are reserved for testing.

**Pipe** (Li et al., 2022a): This benchmark aims to estimate the horizontal fluid velocity based on the structure of the pipe. The governing equations are as follows:

$$\nabla \cdot \mathbf{U} = 0, \tag{96}$$

$$\frac{\partial \mathbf{U}}{\partial t} + \mathbf{U} \cdot \nabla \mathbf{U} = \mathbf{f}^{-1} \frac{1}{\rho} \nabla p + \nu \nabla^2 \mathbf{U}. \tag{97}$$

Each sample represents the pipe as a structured mesh with dimensions $129 \times 129$. The input tensor, shaped as $129 \times 129 \times 2$, encodes the position of each discretized mesh point. The output tensor, with a shape of $129 \times 129 \times 1$, provides the velocity value at each point. For training, 1000 samples with varying pipe shapes are generated, while 200 additional samples, created by altering the pipe's centerline, are reserved for testing.

**Airfoil** (Li et al., 2022a): This benchmark pertains to transonic flow over an airfoil. Due to the negligible viscosity of air, the viscous term $\nu \nabla^2 U$ is omitted from the Navier-Stokes equation. Consequently, the governing equations for this scenario are expressed as follows:

$$\frac{\partial \rho f}{\partial t} + \nabla \cdot (\rho f U) = 0 \tag{98}$$

$$\frac{\partial (\rho f U)}{\partial t} + \nabla \cdot (\rho f U U + p I) = 0 \tag{99}$$

$$\frac{\partial E}{\partial t} + \nabla \cdot ((E + p)U) = 0, \tag{100}$$

where $\rho f$ represents fluid density, and $E$ denotes total energy. The input shape is discretized into a structured mesh with dimensions $221 \times 51$, and the output represents the Mach number at each mesh point. All shapes are derived from the NACA-0012 case provided by the National Advisory Committee for Aeronautics. In the training, 1000 samples from various airfoil designs are used for training, while the remaining 200 samples are reserved for testing.

**Darcy Flow** (Li et al., 2020): This benchmark represents the flow through porous media. 2D Darcy flow over a unit square is given as follows:

$$\nabla \cdot (a(x)\nabla u(x)) = f(x), \quad x \in (0,1)^2, \tag{101}$$

$$u(x) = 0, \quad x \in \partial(0,1)^2. \tag{102}$$

where $a(x)$ is the viscosity, $f(x)$ is the forcing term, and $u(x)$ is the solution. This dataset employs a constant value of forcing term $F(x) = \beta$. Further, Equation 101 is modified in the form of a temporal evolution as follows:

$$\partial_t u(x,t) - \nabla \cdot (a(x)\nabla u(x,t)) = f(x), \quad x \in (0,1)^2, \tag{103}$$

In this dataset, the input is represented by the parameter $a$, and the corresponding output is the solution $u$. The process is discretized into $421 \times 421$ regular grid and then downsampled to a resolution of $85 \times 85$ for main experiments. For training, 1000 samples are used, 200 samples are generated for testing, and different cases contain different medium structures.

## D. Implementation Details

This section presents a comprehensive overview of the experimental setup, covering benchmark datasets, evaluation metrics, and implementation details to ensure a rigorous and reproducible analysis.

*Table 6.* Training and model configurations of LaMO. Training configurations are directly from previous works without extra tuning. For the Darcy dataset, we adopt an additional spatial gradient regularization term $l_{\text{gdl}}$ following ONO work.

| CONFIGURATION | BENCHMARKS | | | | | |
|---|---|---|---|---|---|---|
| | DARCY | NAVIER–STOKES | ELASTICITY | PLASTICITY | AIRFOIL | PIPE |
| **TRAINING** | | | | | | |
| LOSS FUNCTION | $l_2 + 0.1 l_{\text{gdl}}$ | RELATIVE $l_2$ | | | | |
| EPOCHS | 500 | | | | | |
| INITIAL LR | $5 \times 10^{-4}$ | | $10^{-3}$ | | | |
| OPTIMIZER | ADAMW | | | | | |
| BATCH SIZE | 4 | 2 | 1 | 8 | 4 | 4 |
| SCHEDULER | ONECYCLELR | | | | | |
| **ARCHITECTURE** | | | | | | |
| LAYERS | 8 | | | | | |
| EMBEDDING DIM | 64 | 256 | 128 | | | |
| LATENT TOKENS | 1936 | 1024 | 64 | | | |
| DSTATE | 64 | | | | | |
| HEADS | 1 | 4 | | | | |

## D.1. Training Detail

As detailed in Table 5 and Table 6, the datasets were split into training and testing sets by configurations adopted from prior works. Each dataset corresponds to a specific task and follows these established settings. We employed the Adam optimizer with the OneCycleLR scheduler to optimize model performance for training. The relative $l_2$ error metric evaluated and reported results across all benchmarks. Further specifics on the dataset splits, tasks, and training configurations are available in the respective tables for additional clarity for each benchmark dataset.

## D.2. Hyperparameters Details

Table 6 shows that all baselines and benchmarks were trained using consistent configurations, ensuring that LaMO maintains fewer or comparable parameters relative to transformer-based baselines. The table provides an overview of the training and model configurations for LaMO across multiple benchmark datasets. The training configurations include the use of a relative $l_2$ loss term across all datasets, with an additional spatial gradient regularization term $l_{\text{gdl}}$ incorporated for the Darcy dataset as $l_2 + 0.1 l_{\text{gdl}}$ following ONO (Xiao et al., 2023). The model is trained for 500 epochs using the AdamW optimizer, with a OneCycleLR scheduler employed for learning rate adjustment.

On the architectural details of the LaMO, all benchmarks utilize an 8-layer latent architecture, with a fixed state dimension (*DState*) 64 and a convolutional kernel size of 3 across all datasets. SSM heads are configured as 1 for Darcy and 4 for the remaining benchmarks. These configurations align with prior works and ensure robust training without additional hyperparameter tuning. For regular grids, a single encoder-decoder pair is used. In contrast, the encoder and decoder share parameters for other cases for irregular mesh and point cloud and are applied to each latent block.

## D.3. Baselines Details

All baseline models have been extensively validated in prior work. For our experiments, we utilized the official codebase for each operator, ensuring all settings were implemented as prescribed in prior work.

**Typical Neural Operator:** We rely on widely recognized baselines, reporting performance metrics for most operators based on the results provided in their respective official papers for fair comparison. For models that cannot directly handle irregular datasets, we applied the transformation method outlined in the GeoFNO paper to evaluate and report their performance on the corresponding benchmarks.

**Transformer Neural Operator:** For GNOT, OFormer, and ONO, the performance metrics reported on the benchmark were

---

**Algorithm 1** Latent SSM (Multidirectional Latent SSM)

1: **Input:** Latent tokens $\mathbf{Z}_{l-1} \in \mathbb{R}^{B \times M \times D}$
2: $\mathbf{Z}_{l-1} \leftarrow \text{LayerNorm}(\mathbf{Z}_{l-1})$
3: $\hat{\mathbf{Z}} \leftarrow \text{Linear}(\mathbf{Z}_{l-1})$
4: $\hat{\mathbf{X}} \leftarrow \text{Linear}(\mathbf{Z}_{l-1})$
5: $\Delta, B, C \leftarrow \text{Linear}(\hat{\mathbf{X}})$
6: $\bar{\mathbf{A}}, \bar{\mathbf{B}} \leftarrow \text{Discretization}(\Delta, A, B)$
7: $\mathbf{Y} \leftarrow 0$
8: **for** d **in** Direction-Scan **do**
9: $\quad \mathbf{Y} \mathrel{+}= \text{Multihead-SSM}(\hat{\mathbf{X}}_{\text{d}})$
10: **end for**
11: $\mathbf{Y} \leftarrow \mathbf{Y} \odot \text{Activation}(\hat{\mathbf{Z}})$
12: $\mathbf{Z}_l \leftarrow \text{Linear}(\mathbf{Y})$
13: **Output:** Latent tokens $\mathbf{Z}_l \in \mathbb{R}^{B \times M \times D}$

---

directly taken from their respective official publications. In the case of Transolver, recognized as the state-of-the-art (SOTA) operator, we conducted additional evaluations by running the official codebase three times. This ensured consistency and accuracy in reporting the benchmark results. We conducted experiments for the remaining baseline models using their official codebases on GitHub. For reproducibility, we have provided our implementation codebase as a supplementary.

### D.4. Evaluation Metric

Our experimental evaluation focuses on standard PDE benchmarks, utilizing the mean relative $\ell_2$ error (Li et al., 2020) as the primary metric to assess the accuracy of the predicted physics fields. This metric is consistently reported across all experiments and is defined as follows:

$$\mathcal{L} = \frac{1}{N} \sum_{i=1}^{N} \frac{\|\mathcal{G}_\theta(a_i) - \mathcal{G}^\dagger(a_i)\|_2}{\|\mathcal{G}^\dagger(a_i)\|_2}, \tag{104}$$

where $N$ represents the number of samples, $\mathcal{G}_\theta(a_i)$ denotes the predicted solution, and $\mathcal{G}^\dagger(a_i)$ is the ground truth. The inclusion of the normalizing term $\|\mathcal{G}^\dagger(a_i)\|_2$ ensures the metric accounts for variations in absolute resolution scales across different benchmarks, enhancing comparability.

### D.5. Algorithm

In this subsection, we introduce the latent SSM algorithms. In Algorithm 1, the direction scan refers to the axis along which the SSM operates. Experimentally, we employ a bidirectional scan for irregular mesh data. At the same time, for regular grids, we utilize a multidirectional scan as shown in Figure 6 along four paths: (i) top-left to bottom-right, (ii) bottom-right to top-left, (iii) top-right to bottom-left, and (iv) bottom-left to top-right. Each direction is processed in parallel, reducing the overall computational complexity to linear. The multidirectional scan ensures a non-causal kernel, enhancing the model's ability to capture the underlying latent dynamics more effectively.

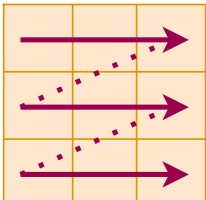 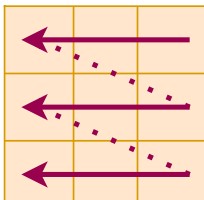 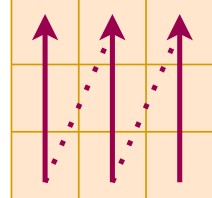 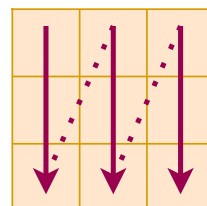

*Figure 6.* Visualization of multidirectional scan for regular grid benchmark dataset.

# E. Additional Experiments

This section presents additional experimental results to evaluate our proposed method comprehensively. This includes ablation studies to assess the impact of key components, scaling experiments to analyze performance across different problem sizes, standard deviation analysis to quantify robustness, and interpretability studies for deeper insights.

## E.1. Ablations

**Ablation on Bidirectional SSM and Weight Sharing**: Table 7 demonstrates the impact of different scanning strategies on LaMO's performance across various datasets. The unidirectional scan consistently underperforms compared to the bidirectional setting, emphasizing the necessity of a non-causal bidirectional scan. This is particularly evident in the Airfoil dataset, where the error worsens from $0.48 \rightarrow 0.91$. Furthermore, employing a multidirectional scan leads to further performance gains for regular grid data, underscoring the advantages of incorporating directional diversity (for Navier Stokes, it improves from $0.10 \rightarrow 0.04$). Additionally, sharing encoder and decoder parameters across blocks effectively reduces model complexity while maintaining competitive performance, as seen in the improved results for Airfoil and Elasticity and comparable accuracy in Plasticity and Pipe.

*Table 7.* Comparison of different LaMO configurations with different scanning directions and weight sharing across all the benchmarks. w/ denotes the with, and w/o denotes the without settings. Lower relative $l_2$ error indicates better model performance.

| OPERATOR ($\times 10^{-2}$) | POINT CLOUD | REGULAR GRID | | STRUCTURED MESH | | |
| --- | --- | --- | --- | --- | --- | --- |
| | ELASTICITY | NAVIER–STOKES | DARCY | PLASTICITY | AIRFOIL | PIPE |
| LaMO (UNIDIRECTIONAL) | $0.97_{\pm 0.044}$ | $16.24_{\pm 0.320}$ | $1.33_{\pm 0.033}$ | $0.22_{\pm 0.022}$ | $0.91_{\pm 0.028}$ | $0.88_{\pm 0.041}$ |
| LaMO (BIDIRECTIONAL W/O WEIGHTS SHARED) | $0.54_{\pm 0.026}$ | $10.1_{\pm 0.252}$ | $0.64_{\pm 0.022}$ | $\mathbf{0.07}_{\pm 0.008}$ | $0.48_{\pm 0.018}$ | $\mathbf{0.38}_{\pm 0.023}$ |
| LaMO (BIDIRECTIONAL W/ WEIGHTS SHARED) | $\mathbf{0.50}_{\pm 0.021}$ | $12.2_{\pm 0.108}$ | $0.77_{\pm 0.015}$ | $0.08_{\pm 0.01}$ | $\mathbf{0.41}_{\pm 0.013}$ | $0.41_{\pm 0.027}$ |
| LaMO (MULTIDIRECTIONAL) | $0.62_{\pm 0.030}$ | $\mathbf{4.60}_{\pm 0.108}$ | $\mathbf{0.39}_{\pm 0.015}$ | $0.12_{\pm 0.014}$ | $0.50_{\pm 0.016}$ | $0.48_{\pm 0.031}$ |

## E.2. Standard Deviations

In this section, we report the standard deviation of all benchmarks while presenting the mean values in the main results table. We repeat all experiments across multiple independent runs and compute the standard deviation to ensure statistical rigor. Table 8 shows that LaMO consistently outperforms all baselines within a 95% confidence interval. We also run the second-best baselines three times for a fair comparison and report their standard deviations. Given the variability in their results across different benchmarks, achieving superior performance across all previous models is challenging. This consistency further validates the robustness and effectiveness of our proposed model.

*Table 8.* The standard deviations of LaMO are reported across all experiments. For comparison, the performance of the second-best operator is also included. Each experiment was conducted over five independent runs to calculate the standard deviation. Lower relative $l_2$ error indicates better model performance.

| OPERATOR ($\times 10^{-2}$) | POINT CLOUD | REGULAR GRID | | STRUCTURED MESH | | |
| --- | --- | --- | --- | --- | --- | --- |
| | ELASTICITY | NAVIER–STOKES | DARCY | PLASTICITY | AIRFOIL | PIPE |
| SECOND-BEST MODEL | $0.64_{\pm 0.02}$ (TRANSOLVER) | $9.57_{\pm 0.20}$ (TRANSOLVER) | $0.59_{\pm 0.01}$ (TRANSOLVER) | $0.13_{\pm 0.01}$ (TRANSOLVER) | $0.53_{\pm 0.01}$ (TRANSOLVER) | $0.46_{\pm 0.02}$ (TRANSOLVER) |
| LaMO (OURS) | $\mathbf{0.50}_{\pm 0.021}$ | $\mathbf{4.60}_{\pm 0.108}$ | $\mathbf{0.39}_{\pm 0.015}$ | $\mathbf{0.07}_{\pm 0.008}$ | $\mathbf{0.41}_{\pm 0.013}$ | $\mathbf{0.38}_{\pm 0.023}$ |

## E.3. Efficiency

As shown in Figure 7, we compare the training time, inference time, and memory consumption of LaMO against Transolver across all datasets and in Table 9, we compare the model parameter count range across the benchmark compared with baselines. Our results indicate that LaMO demonstrates comparable performance in terms of efficiency metrics. Furthermore, LaMO exhibits significantly better performance across all efficiency metrics for time-dependent PDEs such as Navier-Stokes and Darcy. This advantage makes it well-suited for scaling over regular grids, facilitating the development of foundation

Table 9. Comparison of models' parameters count of LaMO with baselines.

| OPERATOR | FNO | U-FNO | LSM | GNOT | GALERKIN | TRANSOLVER | LaMO |
|---|---|---|---|---|---|---|---|
| PARAMETER(IN M) | 0.9-18.9 | 1.0-19.4 | 4.8-13.9 | 9.0-14.0 | 2.2-2.5 | 2.8-11.2 | 1.1-4.0 |

models for PDEs.

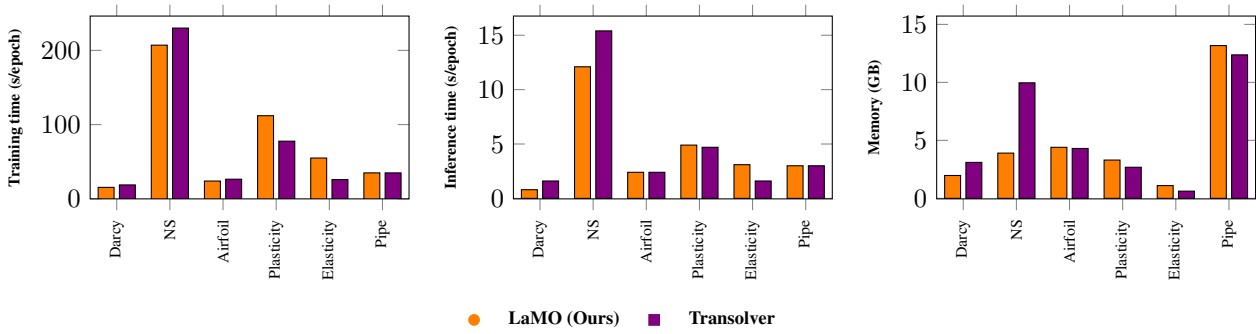

Figure 7. Efficiency comparison between LaMO and Transolver (**Left**) Training Time, (**Middle**) Inference Time and (**Right**) Memory consumption per epoch on all the benchmark dataset.

## E.4. Time Complexity

Table 10 compares the computational complexity of different neural operator models. The models are characterized by their kernel range (Global or Global + Local), kernel type (Fixed or Data-Dependent), and computational complexity for the number of sampling points $N$ in a continuous function. FNO achieves $O(N \log N)$ complexity with a fixed global kernel, Transformers incur $O(N^2)$ complexity due to their global, data-dependent kernel, and LaMO balances efficiency and flexibility with $O(N)$ complexity by leveraging both global and local, data-dependent kernels.

Table 10. Computational complexity of neural operators. $N$ represents the number of sampling points of a continuous function.

| MODEL | KERNEL RANGE | KERNEL TYPE | COMPLEXITY |
|---|---|---|---|
| FNO | GLOBAL | FIXED | $O(N \log N)$ |
| TRANSFORMER | GLOBAL | DATA DEPENDENT | $O(N^2)$ |
| LaMO | GLOBAL + LOCAL | DATA DEPENDENT | $O(N)$ |

## E.5. Model Scalability

From Figure 8, we observe that LaMO consistently outperforms Transolver across all six benchmarks, with a notable error margin in most cases, except for Airfoil, where the improvement is marginal. LaMO achieves a lower relative $l_2$ error than Transolver, with only 40% of the training data for the Darcy and Navier-Stokes benchmarks. In the Plasticity benchmark, LaMO surpasses Transolver with just 20% of the training samples, exhibiting minimal performance degradation despite the reduced dataset size. This is particularly valuable in practical scenarios where collecting large-scale PDE datasets is costly, emphasizing the importance of models that maintain high accuracy with limited training samples.

Furthermore, as presented in Table 11, we evaluate the performance of neural operators in comparison with Transolver across varying network depths on the Navier-Stokes (NS) dataset, which presents significant challenges due to its complex, nonlinear fluid dynamics. Our experiments focus on assessing how the number of layers in the operator architecture influences predictive accuracy. We observe a consistent improvement in performance as the depth of the operator increases, indicating that deeper architectures are better able to capture the intricate temporal and spatial correlations inherent in turbulent flow regimes. Remarkably, even with a relatively shallow configuration of just four layers, LaMO outperforms Transolver, demonstrating its superior efficiency and representational capacity in modeling complex physical systems.

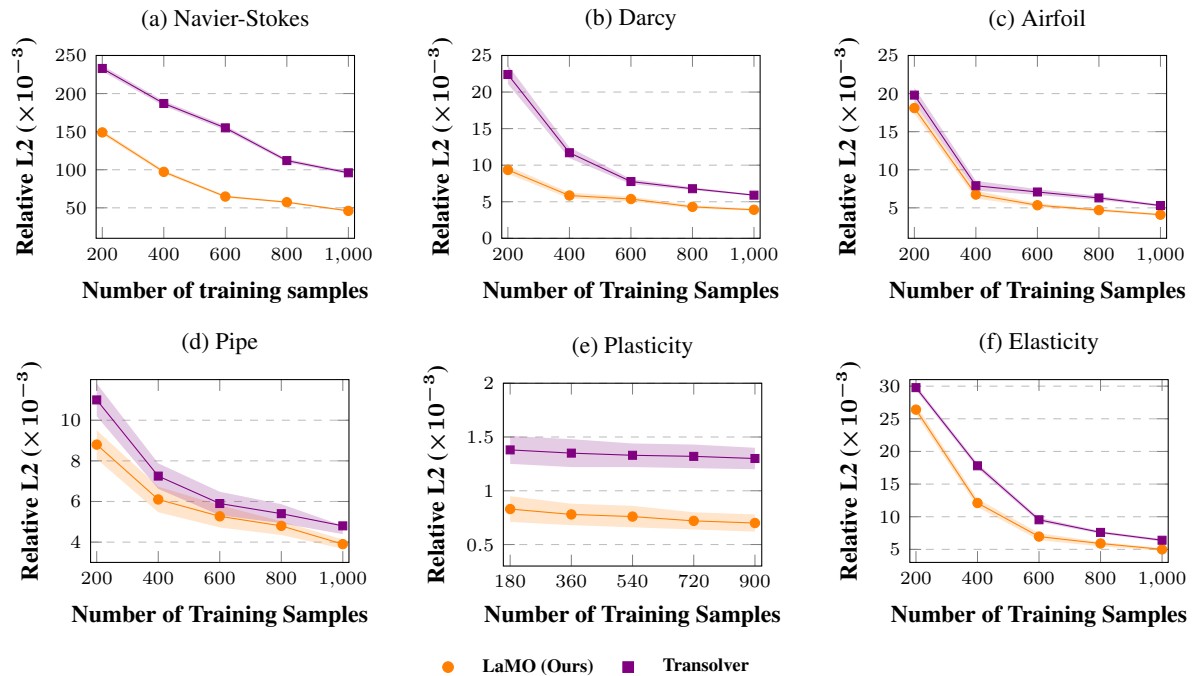

*Figure 8.* Data Efficiency for different benchmarks for LaMO with Transolver (**Top**) (a) Navier Stokes, (b) Darcy (c) Airfoil, (**Buttom**) (d) Pipe, (e) Plasticity and (f) Elasticity benchmark respectively.

*Table 11.* Comparison of relative $l_2$ error on Navier-Stokes benchmark of LaMO with second best baseline model Transolver. Lower relative $l_2$ error indicates better model performance.

| NUMBER OF LAYERS | 2 | 4 | 6 | 8 |
|---|---|---|---|---|
| TRANSOLVER | 0.1601 | 0.1518 | 0.1241 | 0.0957 |
| LAMO | 0.1038 | 0.0608 | 0.0524 | 0.0460 |

### E.6. Interpretability

In Figures 9 and 10, we present the hidden attention matrices of Transolver and LaMO across all layers. For LaMO, we plot the bidirectional attention maps for the Airfoil and Elasticity benchmarks to facilitate an intuitive comparison. As observed, Transolver's attention maps exhibit values concentrated on a few tokens, whereas LaMO's attention is more diffused across tokens, thus establishing a better representation with more diverse latent tokens. The difference can likely be attributed to using softmax in Transolver, which sharpens the attention maps and is known to cause over-smoothing issues.

To further analyze LaMO's performance compared to Transolver, we examine the inter-cosine similarity across layers for both models as shown in Figure 11. LaMO significantly reduces inter-cosine similarity across layers compared to Transolver, indicating better representation learning through SSM. This improvement can be attributed to the absence of the softmax operation, which in Transolver has been identified as causing over-smoothing issues (Ali et al., 2023; Wang et al., 2022).

## F. Visualization

This section presents a comparative analysis of model predictions for Transolver and LaMO across all benchmark datasets. For each dataset, we visualize the predicted outputs alongside their corresponding ground truth values to assess the performance of both models. Additionally, we include heatmaps to highlight the spatial distribution of errors and deviations, providing deeper insights into the strengths and weaknesses of each approach. These visualizations demonstrate the effectiveness of LaMO in capturing intricate patterns within the data while benchmarking it against the existing state-of-the-art model, Transolver.

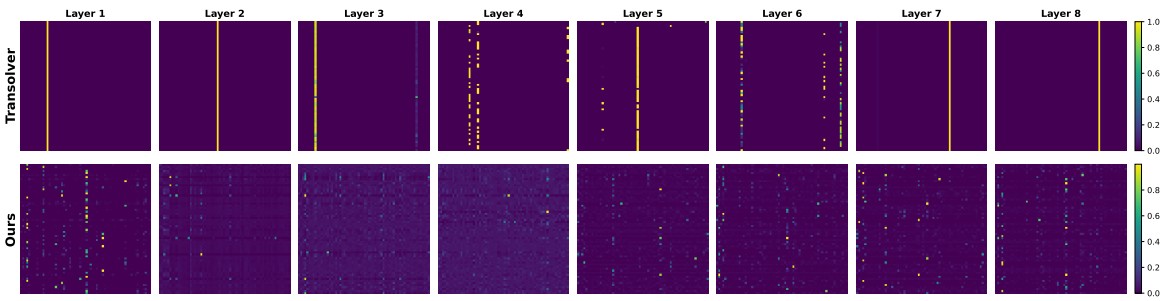

*Figure 9.* Hidden Attention Matrices comparison of Transolver (**Top**) with LaMO (**Bottom**) across the layers on Airfoil benchmark.

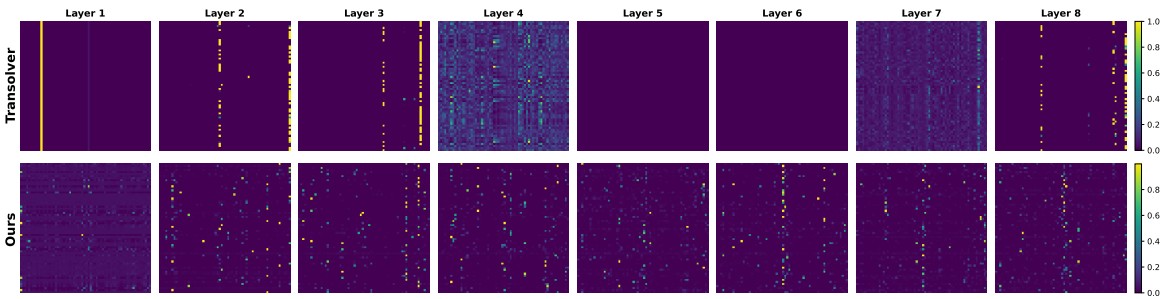

*Figure 10.* Hidden Attention Matrices comparison of Transolver (**Top**) with LaMO (**Bottom**) across the layers on Elasticity benchmark.

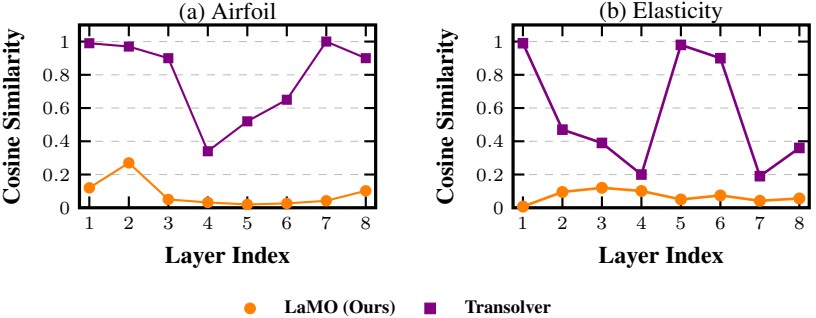

*Figure 11.* Intercosine similarity of tokens across layers for LaMO and Transolver on (a) Airfoil and (b) Elasticity benchmark, respectively.

## G. Limitations and Future Work

LaMO demonstrates promising results in solving parametric PDEs, but its efficiency in an unsupervised setting remains un-explored. Investigating unsupervised training methods and improved token scanning techniques could enhance performance. Additionally, evaluating LaMO's scalability and optimizing its training strategies are key areas for research. For future work, we plan to investigate the potential of using SSM-based operators as foundation models for more efficient PDE solving, focusing on unsupervised learning to achieve better representations. However, the compatibility of existing pretraining methods with SSM-based architectures for operators and developing pretraining techniques specifically tailored for these models remains an open area of exploration. Additionally, we aim to explore improved training strategies for SSM-based operators, including enhanced scanning methods for regular grid PDEs, to optimize their performance further.

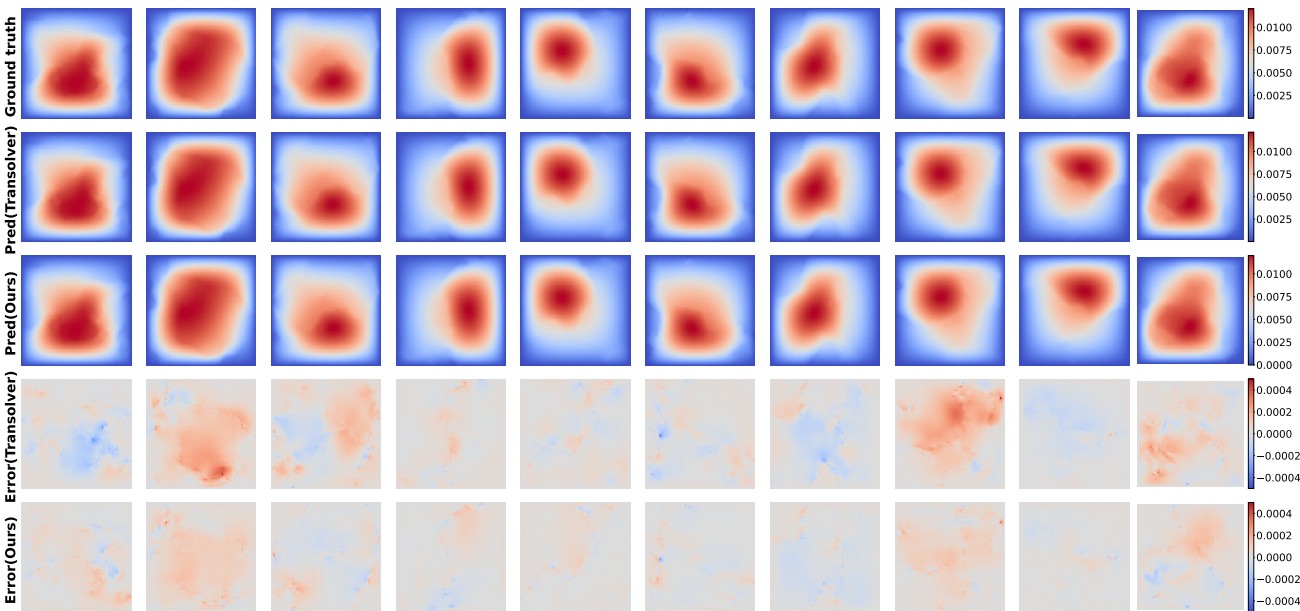

*Figure 12.* **Model Prediction Comparison:** The figure compares Transolver and LaMO on the Darcy dataset. The (**Top**) row represents the ground truth, the (**Middle**) row shows the predictions from Transolver and LaMO, and the (**Bottom**) row illustrates the error heatmap, capturing the differences between the ground truth and the predicted results.

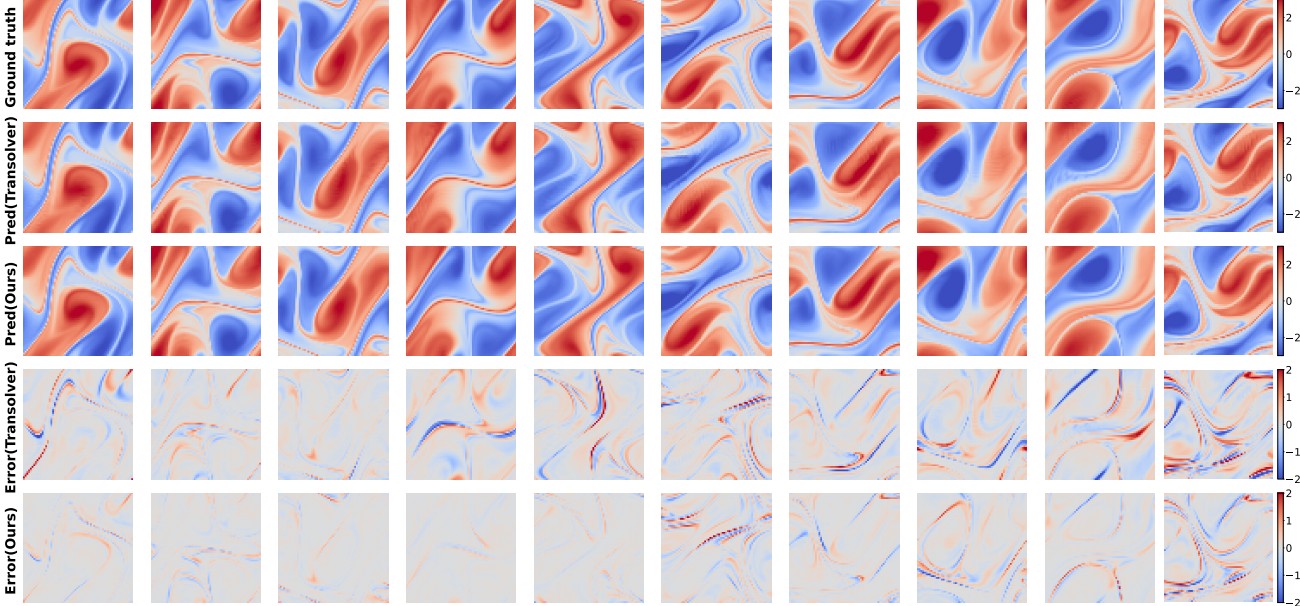

*Figure 13.* **Model Prediction Comparison:** The figure compares Transolver and LaMO on the Navier Stokes dataset. The (**Top**) row represents the ground truth, the (**Middle**) row shows the predictions from Transolver and LaMO, and the (**Bottom**) row illustrates the error heatmap, capturing the differences between the ground truth and the predicted results.

