# OpenReview forum: "Latent Mamba Operator for Partial Differential Equations"
_ICML.cc/2025/Conference — ICML 2025 poster_

### Official Review · Reviewer_hGk2 · 2025-02-20

**Overall Recommendation:** 3

**Summary:**

This paper introduces LaMO, which is an SSM-based neural operator designed to overcome the computational limitations of traditional neural operators for solving PDEs. It establishes a kernel integral interpretation of the SSM framework, proving its equivalence to integral kernel neural operators. It achieves an average 32.3% improvement over existing neural operators across multiple PDE benchmarks, including Navier–Stokes, Darcy flow, and elasticity problems.

## update after rebuttal
Thank you for addressing the concerns. I am keeping my original score.

**Claims And Evidence:**

- Weaknesses:
  - Computational complexity analysis: While LaMO is claimed to be more efficient, a more precise breakdown of runtime (e.g., FLOPs, GPU memory usage per operator) would improve the claims.
  - Ablation studies on kernel choices: The study does not fully explore different kernel configurations for SSM parameterization.

**Essential References Not Discussed:**

N/A

**Experimental Designs Or Analyses:**

- Weaknesses:
  - Limited discussion on hyperparameter selection: The role of latent dimension choices, SSM state sizes, and discretization steps could be better explained.
  - More runtime benchmarks needed: While Figure 4 suggests efficiency, a breakdown of training vs. inference time cost would be helpful.

**Methods And Evaluation Criteria:**

- Strengths: Multiple benchmark PDEs and baselines are tested. The evaluation metric (relative L2) also complies with standard practice.
- Weaknesses: While the benchmarks are diverse, an application to real-world turbulent fluid dynamics (e.g., weather modeling, aerodynamics) would strengthen the evaluation.

**Other Comments Or Suggestions:**

Impressive work! Besides the theoretical contributions, this work adds a novel type of kernel operators (Mamba) to the family of neural operators for PDEs. Its experiments also cover many SOTA baselines, providing a good benchmarking framework for future work on neural operators and Mamba-like kernels. The community would appreciate it if the authors make their code publicly available. (I have gone through the Supplementary Material.) I believe the authors will, yeah^^?

**Other Strengths And Weaknesses:**

N/A

**Questions For Authors:**

See the "Weaknesses" items above.

**Relation To Broader Scientific Literature:**

Sparse kernel methods (e.g., Gaussian Processes for PDEs) could be referenced for comparison.

**Theoretical Claims:**

Claims look good in general. Details of math are not checked line by line.

---

> ### Author Rebuttal · Authors · 2025-03-31
>
> Thank you for the positive comments. Please see the responses to your questions below.
>
> >Computational complexity analysis: While LaMO is claimed to be more efficient, a more precise breakdown of runtime (e.g., FLOPs, GPU memory usage per operator) would improve the claims.
>
> **A:** The above response in Table 3 presents the range of parameters per operator. Additionally, Figure 4 in the main text and Appendix Section E.3 offer a detailed analysis of the training, inference, and memory consumption, providing a precise breakdown of the computational requirements.
>
> >Ablation studies on kernel choices: The study does not fully explore different kernel configurations for SSM parameterization.
>
> **A:** In our experiments, we primarily investigate the effect of directionality in kernel parameterization, as presented in Table 6 of the Appendix. The kernel formulation is inspired by Mamba, which employs time-variant parameterization for matrices B and C. Additionally, matrix A follows a diagonal structure, demonstrating superior performance compared to time-invariant parameterization in computer vision and LLM tasks.
>
> >Weaknesses: While the benchmarks are diverse, an application to real-world turbulent fluid dynamics (e.g., weather modeling, aerodynamics) would strengthen the evaluation.
>
> **A:** In response to your suggestion, we have conducted additional experiments on the ERA dataset, which is derived from the fifth generation of ECMWF reanalysis data. Following standard weather forecasting practices, we selected the geopotential height at 500 hPa (Z500) with a resolution of 2.5° and a time interval of 3 hours as our data. The objective is to predict the geopotential height for 10 timesteps given 2 observations. Due to time constraints and the computational limitation of the experiments, we adopted a data split of 1000/200, consistent with the NS benchmark. We utilized 4 layers for the architecture for both Transolver and LaMO models. Our results indicate that LaMO outperforms Transolver by 25%, demonstrating its superior accuracy in this setting. We plan to explore this direction further as part of our future work.
>
>
> | Operator |  Train error | Test error |
> | --------------- | --------     | --------   |
> | Transolver      | 0.04197      | 0.31665    |
> | LaMO            | 0.04072      | 0.23746    |
>
> ***Table 6: Relative L2 error on ERAZ500 dataset.***
>
> >Limited discussion on hyperparameter selection: The role of latent dimension choices, SSM state sizes, and discretization steps could be better explained.
>
> **A:**  We have employed a learning rate selected from the set {1e-3, 5e-4, 1e-4}, while the remaining hyperparameters are consistent with those used in Transolver. For the SSM, we utilized a state dim (DState) of 64 and varied the number of heads in the range {1, 4}.
>
> The influence of the latent dim and SSM state size, which follow the same configuration as Mamba, is presented in Table 2 of the main text and Table 7 in the Appendix. Our ablation indicates an optimal latent dimension, beyond which performance decreases before increasing again. Regarding discretization, we employed the ZOH scheme, which is similar to Mamba.
>
> >More runtime benchmarks needed: While Figure 4 suggests efficiency, a breakdown of training vs. inference time cost would be helpful.
>
> **A:** We have included the efficiency analysis, covering training time, inference time, and memory consumption, in Appendix Section E.3, where we compare our method with Transolver.
>
> >Sparse kernel methods (e.g., Gaussian Processes for PDEs) could be referenced for comparison.
>
> **A:** We highlight that extending sparse kernel methods, such as Gaussian Processes for PDEs, is computationally demanding and may become impractical for high-dimensional or large-scale PDE problems. Additionally, these methods often struggle with capturing complex, non-stationary dynamics, which neural operators are better equipped to handle. We will ensure that the relevant work is appropriately cited in the final version of the manuscript.
>
> >Impressive work! Besides the theoretical contributions, this work adds a novel type of kernel operators (Mamba) to the family of neural operators for PDEs. Its experiments also cover many SOTA baselines, providing a good benchmarking framework for future work on neural operators and Mamba-like kernels. The community would appreciate it if the authors make their code publicly available. (I have gone through the Supplementary Material.) I believe the authors will, yeah^^?
>
> **A:** Yes, we plan to open-source our codebase, which will contribute to benchmarking future operators and fostering further research and development in this area. Please note that our code is already a part of the supplementary material.
>
> With these, we hope to have addressed all the comments. Please let us know if you have any further concerns or questions. If not, we request you support the manuscript by raising the score.

---

### Official Review · Reviewer_ZQ3y · 2025-02-25

**Overall Recommendation:** 3

**Summary:**

- The Latent Mamba Operator (LaMO) is a scalable state-space model integrated with a kernel integral formulation.

- The authors also provide a theoretical foundation for their approach.

- LaMO demonstrates state-of-the-art performance across various problems.

**Claims And Evidence:**

The authors present a novel approach that achieves state-of-the-art performance across different PDEs, conducting extensive experiments on both regular and irregular domains, including turbulence-related problems. They compare their model against multiple baselines, providing a solid empirical foundation. However, additional evaluations on PDEs such as the Poisson equation, Wave equation, and problems involving temporal dynamics would further strengthen their claims.

They also conduct scaling experiments, particularly focusing on the amount of training data. While Figure 2 implicitly provides scaling behavior with model size, a more detailed analysis of how the model scales with parameter count—whether it follows a power law or logarithmic trend—would be valuable.

**Essential References Not Discussed:**

The paper does not mention prior works on SSMs for PDEs, despite their relevance.

Several recent studies explore SSM-based neural operators:

[1] Cheng, C. W., Huang, J., Zhang, Y., Yang, G., Schönlieb, C. B., & Aviles-Rivero, A. I. (2024). Mamba neural operator: Who wins? transformers vs. state-space models for pdes. arXiv preprint arXiv:2410.02113.

[2] Zheng, J., Li, W., Xu, N., Zhu, J., & Zhang, X. (2025). Alias-Free Mamba Neural Operator. Advances in Neural Information Processing Systems, 37, 52962-52995.

[3] Hu, Z., Daryakenari, N. A., Shen, Q., Kawaguchi, K., & Karniadakis, G. E. (2024). State-space models are accurate and efficient neural operators for dynamical systems. arXiv preprint arXiv:2409.03231.


Including discussions on these works would provide a more comprehensive positioning of the proposed approach within the broader landscape of SSM-based PDE solvers.

**Experimental Designs Or Analyses:**

See above

**Methods And Evaluation Criteria:**

The study includes several important ablation experiments, such as the comparison between unidirectional and bidirectional approaches. This is a critical aspect of the analysis. Moreover, the authors investigate resolution invariance, a fundamental property for neural operators, and provide supporting evidence for it.

Additionally, a clearer examination of model scaling with respect to parameter count would enhance the discussion of efficiency.

**Other Comments Or Suggestions:**

I recommend accepting the paper, provided the authors incorporate prior work on SSMs for PDEs. The study is robust and well-executed, with no significant gaps.

**Other Strengths And Weaknesses:**

/

**Questions For Authors:**

- Were the baseline errors sourced from existing literature, or did you train the baseline models yourselves? If you conducted the training, how were the architectures chosen, and what training procedures were followed?

- How does your approach compare to previous works on SSMs for PDEs?

**Relation To Broader Scientific Literature:**

See my comments above.

**Theoretical Claims:**

The paper includes an accessible theoretical study of the proposed model, demonstrating that its SSM formulation approximates a class of integral linear operators.

However, a deeper exploration of the theoretical implications, particularly in comparison to existing SSM-based PDE solvers, would further solidify the claims.

---

> ### Author Rebuttal · Authors · 2025-03-31
>
> Thank you for the positive comments. Please see the responses to your questions below.
>
> >They also conduct scaling experiments, particularly focusing on the amount of training data. While Figure 2 implicitly provides scaling behavior with model size, a more detailed analysis of how the model scales with parameter count—whether it follows a power law or logarithmic trend—would be valuable.
>
> **A:** We provide the parameter count scaling for the Darcy and NS datasets in Table 5. To better illustrate the scaling trend, we plot the parameter count on a log scale against the number of layers. The plot shows that the parameter growth follows a consistent linear trend, confirming the proportional relationship between the layer count and the model complexity.
>
> | # Layers           | 2    | 4    | 8     | 12    | 24    |
> | -------------------------- | ---- | ---- | ----- | ----- | ----- |
> | Navier-Stokes (Transolver) | 2.93 | 5.70 | 11.23 | 16.76 | 33.35 |
> | Navier-Stokes (LaMO)       | 2.72 | 5.17 | 10.06 | 14.95 | 29.62 |
> | Darcy (Transolver)         | 0.74 | 1.43 | 2.82  | 4.21  | 8.38  |
> | Darcy (LaMO)               | 0.38 | 0.63 | 1.14  | 1.65  | 3.18  |
>
> ***Table 5: Parameter count (in M) vs. layer count for NS (ν = 1e-5) and Darcy.***
>
> >Additionally, a clearer examination of model scaling with respect to parameter count would enhance the discussion of efficiency.
>
> **A:** Please refer to Table 2, Table 4, and Figure 2 in the main text for details on model scaling concerning parameters and efficiency on the Darcy and Navier-Stokes datasets. As shown, the model performance improves with an increase in the number of parameters (layers).
>
> >However, a deeper exploration of the theoretical implications, particularly in comparison to existing SSM-based PDE solvers, would further solidify the claims.
>
> **A:**  We acknowledge the importance of further exploring the theoretical implications, particularly in comparison with existing SSM-based PDE solvers. We will consider this as part of our future work.
>
> >Several recent studies explore SSM-based neural operators:
>
> **A:**  We have cited [3] under SSMs for PDEs in the related work section. We will ensure that the recent works on SSMs for PDEs [1, 2] are included in the related work section of the final version and a relevant discussion.
>
> >Were the baseline errors sourced from existing literature, or did you train the baseline models yourselves? If you conducted the training, how were the architectures chosen, and what training procedures were followed?
>
> **A:** The baseline errors were sourced from existing literature, except Transolver. Since Transolver represents the SOTA, we ran its official codebase (keeping the hyperparameters, data splits, and other settings the same as the original paper) and reported the best errors from multiple runs. We adhered to the standard training procedures described in Appendix Section C for all experiments.
>
> >How does your approach compare to previous works on SSMs for PDEs?
>
> **A:** Our approach differs from previous works on SSMs for PDEs in several key aspects. While prior studies such as [3] employed unidirectional Mamba models for ODEs (1D baselines), we utilize a latent bidirectional Mamba architecture designed explicitly for PDEs (2D baselines). Furthermore, recent works [1, 2] also adopt unidirectional Mamba models (combined with convolution) but are limited to regular grid PDEs only. In contrast, our method leverages a latent bidirectional Mamba, making it applicable to regular and irregular benchmark datasets. We will include additional discussion in the manuscript to address these points.
>
> With these, we hope to have addressed all your comments. Please let us know if you have any further concerns or questions. If not, we request you support the manuscript by raising the score.

---

### Official Review · Reviewer_9Sjn · 2025-03-12

**Overall Recommendation:** 4

**Summary:**

This paper introduces a new approach to solving PDEs by introducing the SSM-based Neural Operator on the latent space. The proposed method achieves a good balance on performance and efficiency. The authors provide theoretical analysis that reveals the equivalence of LaMO with kernel integration. With extensive experiments, the propsed method outperforms SOTA transformer-based model and achieving comparable computational efficiency.

**Claims And Evidence:**

Results in Supplementary E.3 show that the efficiency of your model is only comparable to TRANSOLVER, but the main text presents it as if your model is superior in all aspects/datasets, which is misleading.

**Essential References Not Discussed:**

Essential references are included.

**Experimental Designs Or Analyses:**

1. The performance of TRANSOLVER on the Pipe dataset differs significantly from the results reported in the original paper. The original result is 0.0033, which is better than yours. Please explain this discrepancy.

2. The effect of the latent encoder/decoder and the SSM has not been quantified, making it hard to distinguish their individual contributions.

**Methods And Evaluation Criteria:**

The proposed methods and evaluation criteria make sense for the application.

**Other Comments Or Suggestions:**

1. In the paragraphs "Latent Tokens" and "Efficiency" on Page 8, the authors do not indicate which exact table/figure they are referring to.

2. In the analysis of computational complexity (Page 5 Computational Analysis and Supplementary E.4.), while M is theoreticallt constant, but in practice it is a hyperparameter and will be set due to the size of N and other factors. You can compare the size of M and logN to give readers a better sense of the complexity of your model.

**Other Strengths And Weaknesses:**

Strengths:

1. The paper is well written.
2. Comprehensive experiments, outperforming SOTA
3. Establish a connection between SSMs and kernel integral operators

**Questions For Authors:**

NA

**Relation To Broader Scientific Literature:**

They used Mamba (Gu & Dao, 2023) as the key component of the model and their Latent Encoder is inspired by the Perceiver (Jaegle et al., 2021).

**Theoretical Claims:**

I have checked all the proofs and found no obvious issues.

---

> ### Author Rebuttal · Authors · 2025-03-31
>
> Thank you for the positive comments and for supporting the work. Please see the responses to your questions below.
>
> >Results in Supplementary E.3 show that the efficiency of your model is only comparable to TRANSOLVER, but the main text presents it as if your model is superior in all aspects/datasets, which is misleading.
>
> **A:** Thank you for pointing this out. We agree that LaMO's efficiency is indeed comparable to that of TRANSOLVER. However, in the main text, our reference to efficiency pertains specifically to comparisons with transformer-based baselines such as GNOT and ONO, where LaMO demonstrates superior efficiency. We will revise the final version's text to clarify this distinction and avoid any potential misinterpretation.
>
> >The performance of TRANSOLVER on the Pipe dataset differs significantly from the results reported in the original paper. The original result is 0.0033, which is better than yours. Please explain this discrepancy.
>
> **A:** All the baseline errors were sourced from existing literature, except Transolver. Since Transolver represents the SOTA, we ran its official codebase (keeping the hyperparameters, data splits, and other settings the same as the original paper) and reported the best errors from multiple runs in our manuscript. However, despite our best efforts, we could not replicate the result of 0.0033 reported in the original paper. This discrepancy may be due to differences in the computing environment or random initialization settings. To demonstrate this further, we have made the log file for multiple runs on the pipe dataset available on our anonymous GitHub repository. We shall clarify this in the final version of the paper.
>
> >The effect of the latent encoder/decoder and the SSM has not been quantified, making it hard to distinguish their individual contributions.
>
> **A:** Following your suggestion, in Table 4 (below), we present an ablation study on the latent encoder/decoder, where we observe that ViT patches outperform the Perceiver encoder on regular grids. Additionally, the effects of the SSM and the encoder are detailed in Table 2 of the main text and further elaborated in Appendix Section E.1.
>
>
> | Encoder/Decoder type | Relative L2 |
> | -------------------- | ----------- |
> | ViT                  | 0.0039      |
> | Perceiver (Unshared) | 0.0051      |
> | Perceiver (Shared)   | 0.0056      |
>
> ***Table 4: Ablation on effect of latent encoder/decoder on Darcy.***
>
> >In the paragraphs "Latent Tokens" and "Efficiency" on Page 8, the authors do not indicate which exact table/figure they are referring to.
>
> **A:** In the "Efficiency" paragraph, we are referring to Figure 4 in the main text. However, in the "Latent Tokens" paragraph, we are not referencing any specific table or figure, as it presents an additional ablation study on using latent tokens on a regular grid for the foundation model.
>
> >In the analysis of computational complexity (Page 5 Computational Analysis and Supplementary E.4.), while M is theoreticallt constant, but in practice it is a hyperparameter and will be set due to the size of N and other factors. You can compare the size of M and logN to give readers a better sense of the complexity of your model.
>
> **A:** We acknowledge that while $M$ is theoretically constant, it functions as a hyperparameter in practice, influenced by the size of $N$ and other factors. We will enhance the discussion by comparing $M$ to $log N$ better to illustrate its impact on the model's computational complexity.
>
> We have addressed all your concerns. Please let us know if you have any further comments or questions.

---

### Official Review · Reviewer_SQnF · 2025-03-12

**Overall Recommendation:** 4

**Summary:**

This paper introduces a methodology to use Mamba based architecture to model the spatial dynamics of a PDE, in an operator based setting. The authors take inspiration from the Perceive model and use a latent space, in addition to directly modeling the input physical properties, however instead of a Transformer layer, the authors use a Mamba layer.

The key motivation here it to reduce the quadratic complexity of Transformer to that a linear complexity incase of Mamba. Furthermore, the hope is that Mamba based model is also able to effectively model long range spatial dynamics of the input physical process.

The authors also theoretically prove that in general an SSM based architecture should be able to model the kernel integral operator on a domain, and proving that parametrically at least that the Mamba layer should matches the required structure.

The authors back their claims with extensive empirical analysis, where across a series of benchmarks (like Darcy Flow and 2D Navier-Stokes on regular grids, as well as airfoil, plasticity etc on irregular meshes) Latent Mamba operator is able to outperform baselines such as Transolver.

Strengths:

- The experiments are extensive, and the authors show that their method outperforms the baselines consistently.
- The authors perform ablation studies to explain the importance of different architectural design choices that they make in the design of their architecture.
- They show that their method is efficient and often requires 5-7 times fewer parameters when compared to baselines.

Few Limitations

- They establish that an SSM is a Monte-Carlo Approximation of a Kernel integral operator. However, the theorem does not give any kind of a guarantee in the number of samples (given some assumption on the kernel $\kappa$ (like boundedness, or smoothness assumptions) or the error). I guess the proof is based on the parametric form (Equation 57 of Appendix) of how the SSM approximates the kernel, but even then something that establishes the approximation capacity of the approximation help establish the true equivalence.
    - In general the equivalence is made using $\approx$ which is not formal.
- The scaling results are on Darcy Flow, which potentially tells us nothing. I think similar scaling results should be shown on relatively complex PDEs (perhaps even Navier Stokes).
    - In fact, Darcy Flow is quite a simple dataset that can potentially be learned with very few layers.

Few Suggestions:

- It will be useful to add the number of parameters (or FLOPs) of various methods used in Table 1.

**Claims And Evidence:**

The claim regarding the empirical benefits of the method are appropriately validated through experiments. However, the fact that the SSM layers can approximate the kernel integral operator are lacking given the lack of either number of samples or error bounds.

**Essential References Not Discussed:**

There are a few recent works that try to use SSM based architectures that the authors may have missed:

Hu, Zheyuan, et al. "Deepomamba: State-Space Model for Spatio-Temporal Pde Neural Operator Learning." *Available at SSRN 5149007*.

Ruiz, Ricardo Buitrago, et al. "On the Benefits of Memory for Modeling Time-Dependent PDEs." *arXiv preprint arXiv:2409.02313* (2024).

**Experimental Designs Or Analyses:**

Yes, the empirical results are extensive. The authors show the scaling results on relatively simple datasets on Darcy flow, it will be interesting to see the scaling behaviour on more complex dynamics (such as Navier Stokes).

**Methods And Evaluation Criteria:**

The benchmarks used are on 2D datasets, though it will be interesting to see how the methods perform on 3D. Other than that, the authors also show results on irregular meshes with is good.

**Other Comments Or Suggestions:**

N/A

**Other Strengths And Weaknesses:**

Mostly discussed in the main review.

**Questions For Authors:**

None

**Relation To Broader Scientific Literature:**

Very relevant, given the key point that the quadratic complexity of transformers may be prohibitive for higher dimensions.

**Theoretical Claims:**

There are theoretical claims, though the statement can be made more formal and precise. I think its somewhat straightforward to show though. Please see the main review for more details.

---

> ### Author Rebuttal · Authors · 2025-03-30
>
> Thank you for the positive comments and for supporting the work. Please see the responses to your questions below.
>
> >Sample Complexity: However, the theorem does not...
>
> **A:** A Monte Carlo integral approximation typically has $O\left(\frac{1}{\sqrt{n}}\right)$ convergence rate which is empirically corroborated from Figure 2. We do have a working proof (outlined below) which is subjected to the assumptions stated on $\kappa$. We will be adding it in the supplementary of the manuscript. A formal proof will be explored as part of future work.
>
> **Statement:** A Monte Carlo integral approximation of SSM with suitable assumptions on the kernel (e.g. boundedness and smoothness) provides an upper bound on the error as $\epsilon \sim O\left( \frac{1}{\sqrt{n}} \right)$.
>
> **Proof sketch:**
>
> Boundedness assumption: $\kappa(x, y) \le B$.
>
> In an SSM layer, the approximation of the operator computed via a Monte Carlo sum is: $\[T_n f\](t) = \frac{1}{n} \sum_{i=1}^{n} \varphi(t)^\top A \psi(s_i) f(s_i)$.
>
> Define the error operator as $E_n = T_n - T$.
>
> With probability of at least $1-\delta$ (with $0<\delta<1$), Berstein's inequality suggests that:
>
> $\|T_n - T\| \le C \ \sqrt{\sigma^2 \ \frac{\log(1/\delta)}{n}}$,
>
> where the variance parameter $\sigma^2$ is given by $\sigma^2 = \left\| \sum_{i=1}^{n} \mathbb{E}\left[(Z_i - T)^2\right] \right\|$, and $C$ is a constant.
>
> Hence, to achieve an error of at most $\epsilon$, substituting $\epsilon$ on the LHS, we note that $n \sim O\left( \frac{M}{\epsilon^2} \right)$ where $M$ is constant.
>
> Therefore, $\epsilon \sim O\left( \frac{1}{\sqrt{n}} \right)$.
>
> >Scaling results on relatively complex PDEs (perhaps even Navier Stokes).
>
>  **A:** Following your suggestion, we performed additional experiments to show the scaling results on the Navier-Stokes (NS) dataset in Tables 1 and 2. We note that the operator's performance improves as the number of layers increases on the NS dataset. Notably, even with only 4 layers, LaMO achieves superior performance compared to Transolver (see Tab. 1). Furthermore, Tab. 2 demonstrates that LaMO trained on 400 NS samples delivers a performance comparable to that of Transolver trained on 1000 samples, highlighting LaMO's efficiency and effectiveness with fewer training samples. For a more comprehensive examination of scalability across different sample sizes, please refer to Appendix Section E.5, which presents the results for all benchmark datasets.
>
> | # Layers | 2    | 4    | 6    | 8    |
> | ---------------- | ---  | ---  | ---  | ---- |
> | Transolver       |0.1601|0.1518|0.1241|0.0957|
> | LaMO (Ours)       |0.1038|0.0608|0.0524|0.0460|
>
> ***Table 1: Relative L2 error vs. layer scaling on NS (ν = 1e-5).***
>
>
> | # Training samples | 200    | 400    |  600 | 800    | 1000   |
> | -------------------------- | ------ | ------ | ---  | ------ | ------ |
> | Transolver                 | 0.2330 | 0.1874 |0.1552| 0.1120 | 0.0957 |
> | LaMO (Ours)                 | 0.1490 | 0.0972 |0.0648| 0.0570 | 0.0460 |
>
>  ***Table 2: Relative L2 error vs. training samples on NS (ν = 1e-5).***
>
>  >Number of parameters (or FLOPs).
>
> **A:** In Table 3, we present the number of parameters for a range of neural operators across all benchmarks. The exact number of parameters used for each dataset is available and will be included in the Appendix. We will incorporate these changes into the manuscript and ensure they are reflected in the final version.
>
> | Operator | FNO      | U-FNO    | LSM      | GNOT    | Galerkin | Transolver | LaMO    |
> | --------------- | -------- | -------- | -------- | --- | -------- | ---------- | ------- |
> | Parameters (M)  | 0.9-18.9 | 1.0-19.4 | 4.8-13.9 | 9-14    | 2.2-2.5  | 2.8-11.2   | 1.1-4.0 |
>
> ***Table 3: Baselines parameter range (in M)***
>
> >Performance on 3D.
>
> **A:** As shown in Table 4 of the Appendix, the benchmark datasets used, such as Navier-Stokes (regular) and Plasticity (point cloud), consist of 2D spatial and 1D temporal dimensions. However, it would be interesting to explore the performance of the methods on datasets with higher physical dimensions (e.g., 3D spatial and 1D temporal) as part of future work.
>
> >Recent literature.
>
> **A:** Thank you for highlighting the relevant references we missed. We will include these recent works on SSMs for PDEs in the related work section of the final version.
>
> We have addressed all your concerns regarding these. Please let us know if you have any further comments or questions.

---

### Decision · Program_Chairs · 2025-05-01

**Decision:**

Accept (poster)

**Comment:**

There is a good consensus for acceptance, despite some concerns, including missed references and small issues in the experimental setting. The reviewers all appreciated the methodological novelty and generalization of MAMBA to learning on function spaces.  recommend the paper for acceptance.

Please incorporate all the additional discussions, missed references and results from the rebuttal and discussions with the reviewers into the final version of the manuscript.

A few references were missed that should be added, and some of the comparisons seemed a little cherry picked, in particular, the discrepancy on the results of the Transolver should be clearly addressed in the final manuscript. Ideally I would recommend the authors reach out to the Transolver authors to try and resolve any potential issue. When comparing the methods in terms of number of parameters, please incorporate the added scaling against the number of layers. It would also be good to look at, or at least discuss Neural Operators methods that are geared towards parameter efficiency, such as the FFNO that is in the table but not the plot, and the Tensorized Fourier Neural Operator (TFNO), that compresses the FNO also presented in the table. The reviewers also asked for more details on the hyper parameters selection, this should be added to the final version.

Similarly, some of the reviewers asked for more complex experiments, which the authors addressed in their rebuttal with an experiment on era. The results in their current state aren’t interpretable: 2.5degrees corresponds to a very small resolution. Why did the authors choose a 3 hour prediction window? Era5 is typically use for 1 or 6hours prediction. Did the authors perform the downsampling themselves to 2.5 degree? If so was that with conservative regrinding? The choice of 1000/200 samples seems arbitrary: how were these chosen? I would expect to use e.g. 5 years for training and an out-of-sample year for validation + 1 for testing. Lastly, please also report the RMSE as is standard for z500 prediction.
These results need further details when added to the paper, including rigorous description of the setting, parameters used, metrics, etc, to ensure reproducibility.

Finally, the authors promised to release the code: please make sure the trained model and scripts to reproduce the new experiments are added to the final version.